# TACKLING THE ABSTRACTION AND REASONING CORPUS WITH VISION TRANSFORMERS: THE IMPORTANCE OF 2D REPRESENTATION, POSITIONS, AND OBJECTS

## ABSTRACT

The Abstraction and Reasoning Corpus (ARC) is a popular benchmark focused on *visual reasoning* in the evaluation of Artificial Intelligence systems. In its original framing, an ARC task requires solving a program synthesis problem over small 2D images using a few input-output training pairs. In this work, we adopt the recently popular *data-driven* approach to the ARC and ask whether a Vision Transformer (ViT) can learn the implicit mapping, from input image to output image, that underlies the task. We show that a ViT—otherwise a state-of-the-art model for images—fails dramatically on most ARC tasks even when trained on one million examples per task. This points to an inherent representational deficiency of the ViT architecture that makes it incapable of uncovering the simple structured mappings underlying the ARC tasks. Building on these insights, we propose VITARC, a ViT-style architecture that unlocks some of the visual reasoning capabilities required by the ARC. Specifically, we use a pixel-level input representation, design a spatially-aware tokenization scheme, and introduce a novel object-based positional encoding that leverages automatic segmentation, among other enhancements. Our task-specific VITARC models achieve a test solve rate close to 100% on more than half of the 400 public ARC tasks strictly through supervised learning from input-output grids. This calls attention to the importance of imbuing the powerful (Vision) Transformer with the correct inductive biases for abstract visual reasoning that are critical even when the training data is plentiful and the mapping is noise-free. Hence, VITARC provides a strong foundation for future research in visual reasoning using transformer-based architectures.

## 1 INTRODUCTION

Developing systems that are capable of performing abstract reasoning has been a long-standing challenge in Artificial Intelligence (AI). Abstract Visual Reasoning (AVR) tasks require AI models to discern patterns and underlying rules within visual content, offering a rigorous test for evaluating AI systems. Unlike other visual reasoning benchmarks such as Visual Question Answering (VQA) (Antol et al., 2015) and Visual Commonsense Reasoning (VCR) (Kahou et al., 2018) that rely on natural language inputs or knowledge of real-world physical properties, AVR tasks do not include any text or background knowledge. Instead, they focus purely on visual abstraction and pattern recognition (Małkiński & Mańdziuk, 2023). One prominent example of AVR is the Abstraction and Reasoning Corpus (ARC) (Chollet, 2019), which is designed to evaluate an AI's capacity for generalization in abstract reasoning. Each ARC task involves transforming input grids into output grids by identifying a hidden mapping often requiring significant reasoning beyond mere pattern matching (cf. Figure 2). While the ARC's original setting is one of few-shot learning, there has been recent interest in studying the ARC in a data-rich setting where task-specific input-output samples can be generated (Hodel, 2024), allowing for the evaluation of deep learning-based solutions.

In this paper, we explore the potential of vision transformers to solve ARC tasks using supervised learning. We assess how well transformers can learn complex mappings for a single task when provided with sufficient training data. Our exploration highlights fundamental representational limitations of vision transformers on the ARC, leading to three high-level findings that we believe provide a strong foundation for future research in visual reasoning using transformer-based architectures:

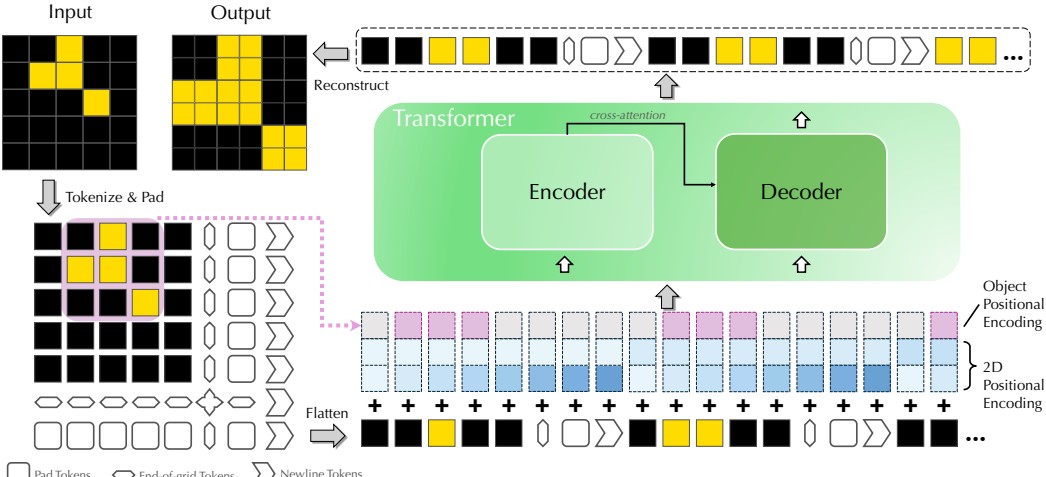

Figure 1: **Overview of our ViTARC framework contribution.** An ARC input image is first tokenized into pixels and padded with visual tokens including end-of-grid tokens that mark the end of the image grid, newline tokens that indicate the end of one row, and pad tokens which are used to pad the image into a fixed maximum size (not drawn in full to maintain clarity). 2D Positional Encodings and Object Positional Encodings are then added to each token before being passed into the transformer. The output tokens are reconstructed into a valid two-dimensional grid.

1. **A vanilla Vision Transformer (ViT) fails on the ARC:** Despite the ARC grids' relatively simple structure compared to the much larger, noisier natural images they are typically evaluated on, a vanilla ViT performs extremely poorly on $90\%$ of the tasks with an overall test accuracy of $18\%$ (cf. Figure 3, Section 3). This is despite using a training set of one million examples per task. Following a failure analysis, we hypothesize that the vanilla ViT fails because it cannot accurately model spatial relationships between the objects in an ARC grid and the grid boundaries.

2. **A 2D visual representation significantly boosts ViT reasoning performance:** Using a 2D representation strategy based on *visual tokens* to represent the ARC input-output pairs, VITARC solves $66\%$ of all test instances – a marked improvement (cf. Section 4). About 10% of the tasks remain poorly solved. After further failure analysis on these tasks, we discover that certain complex visual structures are difficult for VITARC. We hypothesize this is due to limitations of the transformer architecture itself in that it is designed to prioritize token embeddings over positional encodings that can make it challenging to capture intricate spatial relationships.

3. **Positional Information further enhances ViT reasoning abilities:** We improved VITARC's spatial awareness by learning to combine absolute, relative, and *object* positional information (cf. Section 5), resulting in substantial performance boosts, with some ARC tasks progressing from unsolved to fully solved (Figure 3). The final test accuracy is $75\%$, with more than half of the tasks being solved to an accuracy of $95\%$ or more.

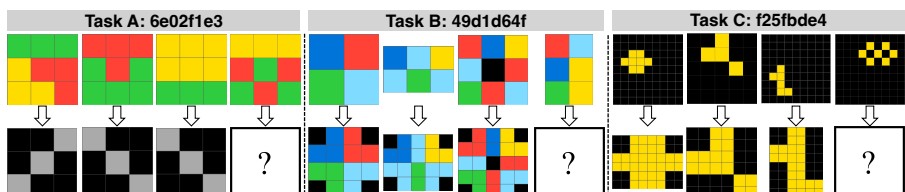

Figure 2: **Three example ARC tasks.** For each task, the first columns contain example input-output pairs from the "training" instances, and the last column contains the "test" instance. The goal is to use the training instances to solve the test instance. The Vanilla ViT setup (Section 3) was only able to solve Task A[1]. Our ViTARC-VT (Section 4) was able to solve Task A and B but still failed at Task C. Our final model ViTARC (Section 5) achieves near 100% accuracy on all three tasks.

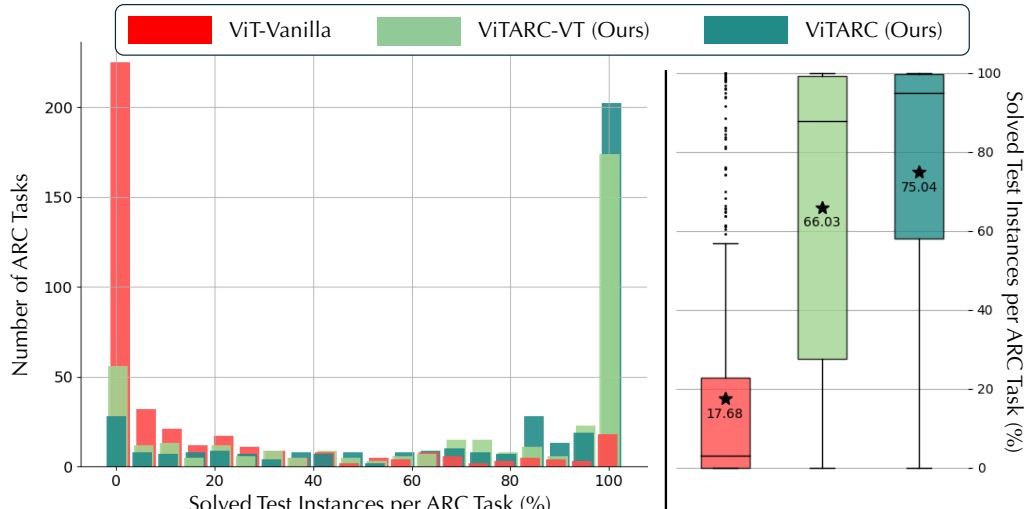

Figure 3: **Model performances on 400 ARC tasks.** Three models are shown: ViT-Vanilla (red) represents the vanilla vision transformer setup (cf. Section 3); ViTARC-VT (light green) and ViTARC (dark green) represent the variants of our framework introduced in Sections 4 and 5, respectively. (Left) Distribution of Solve Rates: The horizontal axis shows the solve rate (percentage of test instances that are solved correctly) on 1000 test instances per task. The vertical axis displays the number of tasks at each solve rate level. (Right) Distribution Statistics: The stars and corresponding values are the overall solve rates across all test instances from all tasks. VITARC-VT and VITARC show significant improvement in performance over the vanilla ViT.

## 2 RELATED WORK

**Abstract Visual Reasoning (AVR)** is an emerging field that seeks to measure machine "intelligence" (Małkiński & Mańdziuk, 2023). Unlike many popular studies that focus on visual reasoning with multi-modal input (Antol et al., 2015; Johnson et al., 2017; Zellers et al., 2019; Bakhtin et al., 2019; Li et al., 2024), AVR focuses on reasoning tasks where the inputs are strictly images. The goal of AVR tasks is to discover abstract visual concepts and apply them to new settings. While the ARC is a generation task using abstract rules, other AVR tasks include classification tasks with explicit rules, such as the Raven's Progressive Matrices (Raven, 2003) and Odd-One-Out (Gardner & Richards, 2006). We refer the readers to Małkiński & Mańdziuk (2023) for a more detailed introduction to AVR.

**Vision Transformers & Positional Encoding.** A Transformer architecture is based on the attention mechanism (Vaswani et al., 2017). Following successes in natural language processing (Brown et al., 2020; Achiam et al., 2023; Devlin et al., 2019), recent studies have extended the Transformer to the vision domain (Han et al., 2023). State-of-the-art approaches involve dividing the image into rectangular "patches"(Dosovitskiy et al., 2021), where various techniques such as dynamic patch sizes allow for more effective capture of local information (Havtorn et al., 2023; Zhou & Zhu, 2023). Vision Transformers have been successfully used to perform various image-to-image generation tasks such as inpainting (Li et al., 2022), image restoration (Liang et al., 2021), colorization (Kumar et al.), and denoising (Wang et al., 2022).

Due to the set-based (permutation-invariant) nature of attention, Positional Encodings are used to inject positional information in a Transformer (Vaswani et al., 2017). State-of-the-art Positional Encodings include Absolute Positional Encodings (APEs) where unique encodings are added to the inputs directly (Devlin et al., 2019), Additive Relative Positional Encodings (RPEs) (Shaw et al.,

---

[1]Task A follows a rule based on color count: if the input grid has two distinct colors, the output contains a grey diagonal from the top-left to the bottom-right. Conversely, if the input grid has three colors, the grey diagonal is from the top-right to the bottom-left.

2018; Raffel et al., 2020; Li et al.) that measure the relative positions between tokens by modifying the attention logits, and various hybrid methods (Su et al., 2024; Zhou et al., 2024). Vision Transformer research has adapted these concepts, implementing both APEs (Dosovitskiy et al., 2021) and RPEs (Wu et al., 2021) to incorporate positional information about the image patches.

**Solvers for the ARC.** Since the introduction of the ARC (Chollet, 2019), the development of solvers has been an active research area. The earliest successful approaches consisted of an expressive Domain Specific Language (DSL) and a program synthesis algorithm that searched for a valid solution program expressed in the DSL. These include DAG-based search (Wind, 2020), graph-based constraint-guided search (Xu et al., 2023), grammatical evolution (Fischer et al., 2020), library learning (Alford et al., 2021), compositional imagination (Assouel et al., 2022), inductive logic programming (Hocquette & Cropper, 2024), decision transformers (Park et al., 2023), generalized planning (Lei et al., 2024), reinforcement learning (Lee et al., 2024), and several others (Ainooson et al., 2023; Ferré, 2021). These models achieved up to 30% on the private ARC test set (Chollet et al., 2020; Lab42, 2023).

Recently, Transformer-based Large Language Models (LLMs) were shown to exhibit an apparent ability to perform "reasoning" (Wei et al., 2022) spurring interest in using LLMs as part of an ARC solver. Such methods were prompted to perform program synthesis on a DSL (Min Tan & Motani, 2024; Barke et al., 2024) as well as general-purpose languages such as Python (Butt et al., 2024; Wang et al., 2024), with the best-performing model achieving 42% on the public ARC evaluation set (Greenblatt, 2024). LLMs were also explored as standalone solvers, where they were asked to produce the output grids directly instead of outputting a program. Although pre-trained LLMs proved ineffective when generating the output grid pixels directly (Camposampiero et al., 2023; Mirchandani et al., 2023; Moskvichev et al., 2023), its performance was shown to be improved by object representation (Xu et al., 2024). The vision variant of a state-of-the-art LLM, GPT-4V was shown to be ineffective (Mitchell et al., 2023; Xu et al., 2024).

The current state-of-the-art solver has achieved 46% on the private test set at the time of writing (arcprize, 2024) but is not publicly available or described in detail. We do know that it is a pre-trained LLM that is fine-tuned on millions of synthetic ARC tasks generated using the RE-ARC generator (Hodel, 2024) and combined with test-time fine-tuning (Cole & Osman, 2023). Despite the visual nature of ARC tasks, Transformer-based LLM approaches convert the images into strings, which does not fully capture all relevant structural information (Xu et al., 2024).

## 3 VANILLA VISION TRANSFORMER FOR THE ARC: AN INITIAL APPROACH

We first implement a vanilla Vision Transformer architecture as detailed in Dosovitskiy et al. (2021) and Touvron et al. (2021) as a solver for the ARC. Consider an input image $I$ divided into $P \times P$ non-overlapping patches. Each patch $p_i$ is flattened in raster order and indexed by $i$ before being projected into a $d$-dimensional embedding space. Let $h_i^0$ denote the initial input to the Transformer for patch $p_i$. For the $n$-th Transformer layer, $n \in \{1, \ldots, N\}$, and for a single attention head, the following operations are performed:

$$h_i^0 = \mathbf{E}_{p_i} + \mathbf{E}_{\text{pos}_i} \tag{1}$$

$$\hat{h}_i^n = \text{LayerNorm}(h_i^{n-1}) \tag{2}$$

$$q_i^n, k_i^n, v_i^n = \hat{h}_i^n \boldsymbol{W}_q^n, \quad \hat{h}_i^n \boldsymbol{W}_k^n, \quad \hat{h}_i^n \boldsymbol{W}_v^n \tag{3}$$

$$A_{i,j}^n = \frac{q_i^n \cdot k_j^n}{\sqrt{d}} \tag{4}$$

$$o_i^n = \sum_j \text{Softmax}(A_{i,j}^n) v_j^n + h_i^{n-1} \tag{5}$$

$$f_i^n = \text{FeedForward}(\text{LayerNorm}(o_i^n)) + o_i^n \tag{6}$$

$$h_i^n = \text{LayerNorm}(f_i^n) \tag{7}$$

Here, $\mathbf{E}_{p_i}$ is the embedding of patch $p_i$ and $\mathbf{E}_{\text{pos}_i}$ is the positional encoding. Following the standard ViT implementation of Dosovitskiy et al. (2021), the Absolute Positional Encoding (APE) is

calculated as a learnable 1D encoding:

$$\mathbf{E}_{\text{pos}_i} = \boldsymbol{W}_i, \quad \mathbf{E}_{\text{pos}_i} \in \mathbb{R}^d, \quad \boldsymbol{W} \in \mathbb{R}^{L \times d}$$

where $\boldsymbol{W}$ is a learned matrix assigning a $d$-dimensional vector to each of the possible $L$ positions; $L$ is the maximum input length.

As seen in Figure 2, ARC tasks are *generative* and require mapping an input image to an output image. Because image dimensions may vary across instances of the same task and even between the input and output grids of the same instance, any model that generates candidate solutions to an ARC input must be able to "reason" at the pixel level. We adapt the ViT architecture to this setting by making the following key modifications:

– We introduce a decoder with cross-attention using the same positional encoding and attention mechanisms of the encoder. After the final decoder layer $N$, the output embedding $h_i^N$ of patch $i$ is projected linearly and a softmax function is applied to predict pixel-wise values $\hat{y}_i$ as $\hat{y}_i = \text{Softmax}(\text{Linear}(h_i^N))$. The cross-entropy loss is computed as the sum over pixels, $-\sum_i y_i \log(\hat{y}_i)$.

– To achieve the required pixel-level precision for the ARC task, we employ a patch size of $1 \times 1$, effectively treating each pixel as an independent input token.

– To handle variable-sized grids, the flattened list of tokens is padded to a fixed maximum length. This configuration enables the model to process and generate ARC task outputs pixel-by-pixel.

### 3.1 EXPERIMENTS

**Data.** To evaluate ViT's reasoning capabilities comprehensively, we treat each of the 400 public training ARC tasks as an individual AVR problem. We generate a dataset of 1 million input-output pairs per task using the RE-ARC generator (Hodel, 2024) and train all of our models (the vanilla ViT and VITARC models) in a supervised manner from scratch.

**Hyperparameters and training protocol.** The ViT baseline consists of three layers with eight attention heads and a hidden dimension of 128. We trained the model on various single-core GPU nodes, including P100, V100, and T4, using a batch size of 8 for one epoch. We chose to train for one epoch because most models showed signs of convergence within the epoch. Due to computational resource limitations, we evaluated our major milestone models on the full set of 400 tasks. However, for the ablation studies hereafter, we used a randomly sampled subset of 100 tasks. For more details on the training process, please refer to Appendix B. Our code is available in the supplementary materials and will be open-sourced upon publication.

**Evaluation metric.** We evaluate the model primarily on the percentage of solved instances, using a strict criterion: an instance is considered solved only if all generated pixels, including padding and border tokens, exactly match the ground truth. This approach is stricter than the original ARC metric which permits up to three candidate solutions.

**Results.** Figure 3 shows that the vanilla ViT performs poorly: a significant percentage of tasks have a near 0% solve rate despite the million training examples per task. This points to fundamental limitations of the ViT architecture that inhibit abstract visual reasoning. In the following sections, we analyze failure cases and investigate methods for enhancing the visual reasoning ability of ViT.

## 4 VISUAL TOKENS: A BETTER REPRESENTATION FOR VIT

The basic version of our VITARC framework builds on the vanilla ViT but includes three simple yet highly effective changes to the representation of the ARC grids. We refer to these changes as *visual tokens* to emphasize a departure from the language-based tokenization perspective in the particular setting of the ARC.

**2D padding.** We observed that a large portion of the incorrect outputs from the vanilla ViT had incorrect grid sizes, a flagrant failure mode; An example is visualized in Figure 4 (ViT-Vanilla). We hypothesize that this is due to the vanilla ViT implementing padding in a "1D" manner, where `<pad>` tokens are applied to the sequence after flattening, thus losing the two-dimensional context. To address this issue, we implemented 2D padding, where `<pad>` tokens are applied to the image *first* before being flattened in raster order into a sequence for transformer processing (see Figure 1).

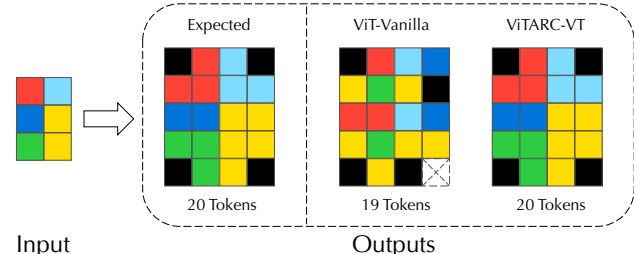

Figure 4: **Visualization of ViT-Vanilla failure case (Task B from fig. 2).** The output of ViT-Vanilla has an incorrect number of tokens (19) compared to the expected 20. For better visualization, the output pixels are arranged to match the grid, although the model generates the pixels in a continuous sequence in raster order. This makes the task a relaxed output prediction for ViT-Vanilla, where the flattened output sequence is compared with the expected output sequence.

However, this design introduces a new drawback: the model must now predict `<pad>` tokens as part of the output grid. In initial experiments, we observed that the model tends to ignore these `<pad>` tokens (that do not receive attention), erroneously predicting over the entire $h_{max} \times w_{max}$ grid rather than focusing on the valid input region. An example of this issue is shown in Figure 8 of Appendix A. To address this, we define `<2d_pad>` tokens and enable attention to these tokens, allowing the model to properly account for the padded regions as well as the valid output region.

**Border tokens for spatial awareness.** The implementation of 2D padding did not completely alleviate the previously observed failure cases. We further observed that for some tasks, when the output is cropped to the true grid dimensions, the predictions within the valid region are correct, underscoring the importance of proper boundary handling. We show an example in Figure 8 of Appendix A. Inspired by the use of end-of-sequence (EOS) tokens like `` in Natural Language Processing (NLP), we introduce *border tokens* to explicitly define the grid boundaries (cf. Figure 1):

- **Newline tokens** (`<2d_nl>`) mark row transitions in the $h_{max} \times w_{max}$ grid.
- **End-of-grid tokens** (`<2d_endxgrid>`, `<2d_endygrid>`, and `<2d_endxygrid>`) delineate the true $h \times w$ grid boundaries.

The introduction of border tokens enables the model to more effectively distinguish the task grid from the padding. Without these tokens, the model would need to count tokens to determine boundaries, which becomes unreliable—especially in ARC tasks with dynamically defined output grid sizes (e.g., task C in Figure 2). Furthermore, as we see in ViT-Vanilla failure cases (Figure 4), it is ambiguous to recover the 2D positions from a 1D sequence of predicted tokens alone. Border tokens also provide a fixed 2D template to fill in, which implicitly helps reconstruct the correct 2D positions and makes it easier to debug the related grid logic.

**2D Absolute Positional Encoding.** With the introduction of 2D padding and border tokens, our setup now operates on fixed-size, two-dimensional input-output pairs that are aligned with a universal $(x, y)$ coordinate system. This allows us to adopt existing positional encoding (PE) strategies from the literature (see Section 2). After empirical analysis, we implement a (non-learned) 2D sinusoidal APE for VITARC, which is defined as follows:

$$\text{Sinusoid}(p) = \begin{bmatrix} \sin\left(\frac{p}{10000^{2k/d}}\right) \\ \cos\left(\frac{p}{10000^{2k/d}}\right) \end{bmatrix}, \quad k = 0, \dots, d/2, \tag{8}$$

$$\mathbf{E}_{\text{pos}_{(x,y)}} = \text{concat}\left(\text{sinusoid}(x), \text{sinusoid}(y)\right), \tag{9}$$

where $p$ represents either the $x$ or $y$ coordinate, $k$ is the index of the positional encoding dimension, and $d$ is the total embedding dimension.

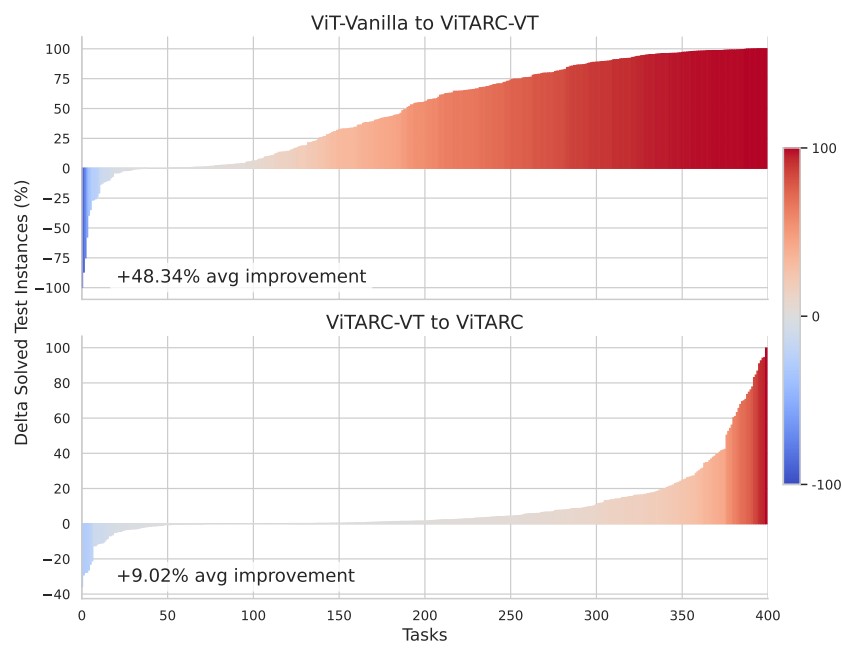

Figure 5: **Improvement in percentage of solved test instances per task.** (a) From ViT-Vanilla to ViTARC-VT: We observe that over 85% of tasks benefit from the introduction of 2D Visual Tokens, showing consistent gains compared to the vanilla ViT. (b) From ViTARC-VT to ViTARC: We observe that more than half of all tasks show further improvement. Improvement from ViT-Vanilla to ViTARC is shown in Figure 9 in Appendix C.1 where a 57.36% average improvement is observed.

## 4.1 RESULTS

Figure 3 shows substantial improvements in test accuracy due to the 2D visual tokens just described. Figure 5(a) illustrates the improvement in the percentage of solved instances for each task. We observe an average performance boost of 48.34% compared to the baseline ViT across the 400 tasks. This model, referred to as ViTARC-VT, demonstrates that the new representation with 2D visual tokens significantly enhances the model's ability to handle AVR tasks.

A key driver of this improvement is the use of 2D padding, which creates a fixed schema for 2D positions. This ensures consistent spatial alignment and effectively addresses the challenge of applying 2DAPE to variable-sized grids, where unknown output positions during inference complicate accurate mapping.

To quantify the contribution of border tokens, we performed an ablation study. As seen in Figure 7, the absence of border tokens leads to a 4.59% decrease in accuracy, emphasizing their importance in helping the model delineate task grid boundaries and maintain spatial consistency in the input representation. For more detailed numerical results, refer to Table 6 in Appendix C.2.

## 4.2 ANALYSIS

While ViTARC-VT delivers strong results—approximately 40% of ARC tasks achieved over 90% solved test instances—there remain certain tasks where the model struggles. Specifically, around 10% of ARC tasks have less than 5% of test instances solved, even after training on a large dataset containing one million examples per task. Closer examination reveals that tasks involving complex visual structures, such as concave shapes, holes, or subgrids, are consistently problematic. These challenges highlight certain architectural limitations, particularly the model's difficulty in segmenting multi-colored objects, where positional information should ideally play a more dominant role.

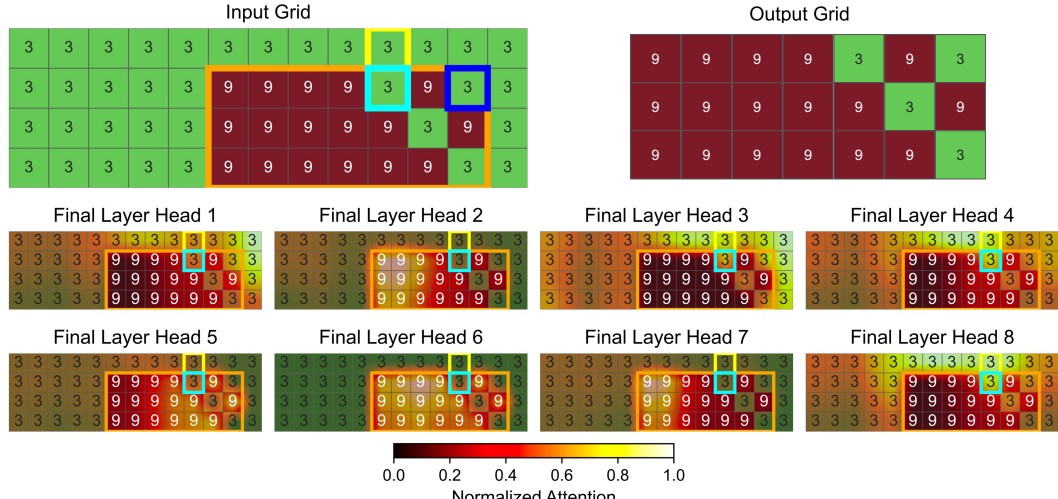

Figure 6: **ViTARC-VT failure analysis for ARC task (#1cf80156).** Cross-attention heatmap across all attention heads in the final layer at the step predicting the color-3 pixel within the dark blue box. The task requires finding the maximum rectangular subgrid in the input. The attention, visualized in a thermal heatmap, shows that none of the heads successfully distinguish the subgrid (orange bounding box) from its surroundings that motivates the PEMixer and OPE, nor do they differentiate the color-3 pixel inside the cyan box (within the subgrid) from the pixel in the yellow box (outside the subgrid) that motivates the 2D-RPE directional bias.

To better understand this behavior, we refer back to Equation (1): $h_i^0 = \mathbf{E}_{p_i} + \mathbf{E}_{\text{pos}_i}$. In this setup, the absolute positional encoding, $\mathbf{E}_{\text{pos}_i}$, is directly added to the input embedding, $\mathbf{E}_{p_i}$, so that it adjusts the token's representation without overwhelming its semantic content. This works effectively in NLP tasks, where the semantic meaning of tokens generally takes precedence over their position. However, in vision tasks, especially those requiring detailed visual reasoning, spatial relationships often carry as much importance as, if not more than, the content of the tokens. For tasks in the ARC that involve complex multi-colored objects, such as subgrids, accurately encoding positional information becomes crucial. Figure 6 illustrates a specific case where the model fails to group pixels within a multi-colored subgrid correctly. The cross-attention map reveals that the model overly relies on color similarity, resulting in confusion between similarly colored pixels in different positions. This indicates a lack of sufficient attention to spatial relationships, which is essential for such tasks and guides us to develop further enhancements in the next section.

## 5    RECENTERING POSITIONS & OBJECTS FOR SPATIAL REASONING IN VIT

Our observations on the failure cases of ViTARC-VT lead us to implement further enhancements to tackle tasks with complex visual structures by better encapsulating the positional information of pixels and objects.

**Positional Encoding Mixer (PEmixer).**    To better balance the importance of positional information and tokens, we modify Equation (1) by learning weight vectors for the encodings, i.e.,

$$h_i^0 = \boldsymbol{\alpha} \odot \mathbf{E}_{p_i} + \boldsymbol{\beta} \odot \mathbf{E}_{\text{pos}_i}, \tag{10}$$

where $\boldsymbol{\alpha}$ and $\boldsymbol{\beta}$ are **learnable** vectors of the same dimension as the encoding vectors, and $\odot$ denotes element-wise multiplication. This effectively allows the model to learn the optimal balance between input tokens and positional encoding.

Furthermore, our implementation of 2D APE as described in Section 4, where $\mathbf{E}_{\text{pos}_{(x,y)}}$ is the concatenation of $\mathbf{E}_{\text{pos}_x}$ and $\mathbf{E}_{\text{pos}_y}$, allows the vector-based mixing coefficients to focus on specific coordinates, which further improves the model's reasoning capability over specific pixels.

**2D Relative Positional Encoding (2D-RPE).** Motivated by the example in Figure 6, we aim to enable the model to distinguish between pixels in different spatial regions, such as the color-3 (green) pixel in the cyan box versus the one in the yellow box. In this example, the positional difference between the two pixels is just 1 along the $y$-coordinate. APE encodes this difference as a small shift; while the transformer is theoretically capable of capturing these spatial relationships, in practice often requires many training epochs (Hahn, 2020).

To better account for spatial relationships in two-dimensional grids, we adapt the Relative Positional Encoding (RPE) approach from ALiBi (Press et al., 2021) and extend it to 2D. ALiBi introduces additive positional biases to the attention scores based on the relative positions of tokens. In its original 1D form, ALiBi defines the positional bias as the following:

$$A_{i,j}^n = \frac{q_i^n \cdot k_j^n}{\sqrt{d}} + \mathbf{B}_{\mathbf{P}_{i,j}}, \quad \mathbf{B}_{\mathbf{P}_{i,j}} = r \cdot |i - j|, \tag{11}$$

where $\mathbf{P}_{i,j}$ represents the relative positional offset between tokens $i$ and $j$, and $r$ is a predefined slope that penalizes tokens based on their distance.

Extending to 2D, we introduce distinct slopes for the "left" and "right" directions, efficiently capturing directional biases along the x and y axes. This design leverages the inherent 2D structure of the data while aligning with the sequential raster order of the generation process. Specifically:

- Pixels located above or to the left of the current pixel in 2D space are assigned a bias $r_{\text{left}}$.
- Pixels located below or to the right are assigned a bias $r_{\text{right}}$.

Hence, the 2D-RPE bias is computed as:

$$\mathbf{B}_{\mathbf{P}_{i,j}} = \begin{cases} r_{\text{left}} \cdot d\left((x_i, y_i), (x_j, y_j)\right), & \text{if } j \leq i, \\ r_{\text{right}} \cdot d\left((x_i, y_i), (x_j, y_j)\right), & \text{if } j > i, \end{cases} \tag{12}$$

where $d\left((x_i, y_i), (x_j, y_j)\right)$ represents the 2D Manhattan distance between coordinates $(x_i, y_i)$ and $(x_j, y_j)$. The slope values $r_{\text{left}}$ and $r_{\text{right}}$ are derived following the ALiBi setup, forming a geometric sequence of the form $2^{-8/n}$ for $n$ heads. $r_{\text{left}}$ starts at $1/2^1$, while $r_{\text{right}}$ starts at $1/2^{0.5}$, both using the same ratio.

In this work, we leverage both 2D-RPE and 2D sinusoidal APE within our model. In contrast to observations made in Swin (Liu et al., 2021), where a degradation in performance was noted when combining RPE with APE, our results demonstrate a marked improvement. The inclusion of 2D-RPE allows for more precise modeling of relative spatial relationships, complementing the global positional information provided by APE. This synergy proves particularly effective for tasks demanding fine-grained spatial reasoning.

**Object-based Positional Encoding (OPE).** For tasks involving multi-colored objects, or more generally, tasks that require objectness priors (Chollet, 2019), external sources of knowledge about object abstractions can be integrated into the model. We inject this information through a novel *object-based positional encoding*. We extend the 2D sinusoidal APE defined in Equation (9) by introducing the object index $o$ as an additional component to the pixel coordinates $(x, y)$. This results in a modified positional encoding:

$$\mathbf{E}_{\text{pos}_{(o,x,y)}} = \text{concat}\left(\text{sinusoid}(o), \text{sinusoid}(x), \text{sinusoid}(y)\right). \tag{13}$$

In object detection models, two primary segmentation methods are bounding box segmentation and instance segmentation, the latter of which captures precise object boundaries. For simplicity, we adopt bounding box segmentation to derive the object index $o$, as fine-grained distinctions at the instance level can already be addressed by the model's attention mechanism, as illustrated in Figure 6. Figure 1 demonstrates how bounding box information is obtained and incorporated into the positional encoding.

This design integrates seamlessly with the PEmixer introduced earlier, as it enables the model to dynamically adjust its reliance on the object index $o$ based on the task's needs. In scenarios where the object index provides valuable abstraction, the model can prioritize it, while in cases where the object-based method is less effective, the model can fall back on the $(x, y)$ positional information.

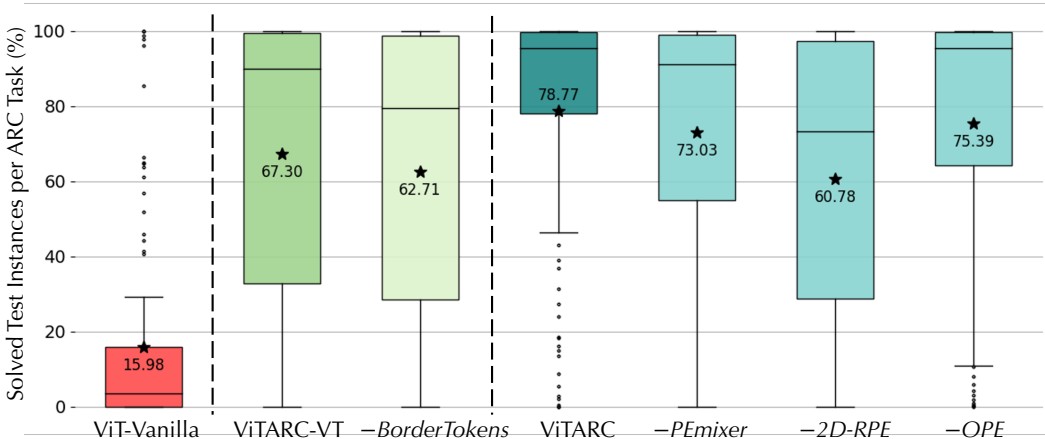

Figure 7: **Distribution statistics of solve rates on 100 random tasks for ablation.** 7 Models are shown: ViT-Vanilla, ViTARC-VT, and ViTARC are the models introduced in Sections 3, 4 and 5 respectively. Ablated components are prefixed as − and ablate the full model to the left, i.e., −*BorderTokens* is an ablation of this component from ViTARC-VT and each of −*PEmixer*, −*2D-RPE*, and −*OPE* ablate these respective components from ViTARC.

For our experiments, OpenCV's contour detection (Bradski, 2000) proved sufficient for generating object indices in the ARC tasks, demonstrating the practical effectiveness of OPE. This novel approach not only addresses challenges related to complex object shapes but also establishes a method for injecting external objectness knowledge into vision models, enhancing their reasoning capabilities.

## 5.1 RESULTS

We arrive at our final model, ViTARC, which contains all the improvements mentioned in Section 4 and Section 5. The final encoding combines all three components: 2DAPE, 2DRPE, and OPE, leveraging their complementary strengths to enhance spatial reasoning. As shown in Figure 3, the model is a significant improvement over both the baseline ViT-Vanilla and ViTARC-VT due to the proposed positional enhancements.

Furthermore, Figure 5(b) highlights the generalization of these improvements across tasks, with an additional 9.02% increase in solved instances compared to ViTARC-VT. ViTARC-VT itself already achieved a significant boost over ViT-Vanilla, culminating in a total improvement of 57.36% over the baseline ViT-Vanilla.

Figure 7 further illustrates the impact of each enhancement on task performance. All three contribute to the overall improvement, with 2D-RPE providing the largest gain, followed by PEmixer and OPE. Notably, without 2D-RPE, the model's performance drops below that of ViTARC-VT. This occurs because OPE, while effective in specific tasks, is not consistently reliable. In these cases, ViTARC must fall back on the $(x, y)$ embeddings from 2D-APE, which are less expressive due to their lower dimensionality compared to ViTARC-VT. The inclusion of 2D-RPE recovers these positional signals at the attention level, ensuring robust performance even when object-based cues are insufficient.

For a comprehensive breakdown of the task-level performance and the numerical details of these ablations, please refer to Appendix C.2.

## 6 CONCLUSION

This paper introduced VITARC, a Vision Transformer architecture designed to address the unique challenges posed by the Abstraction and Reasoning Corpus. A key finding of our work is that positional information plays a critical role in visual reasoning tasks. While often overlooked when adapting transformers from NLP to vision, our results demonstrate that even simple enhancements

to positional encoding can significantly improve performance on ARC tasks. Furthermore, we show that incorporating object indices as additional positional information via OPEs provides a meaning-ful improvement in handling complex spatial relationships in ARC tasks.

Additionally, we introduced 2D padding and border tokens to handle variable-sized images requir-ing high precision in visual reasoning. Given ARC's pixel-level precision and abstract reasoning requirements (e.g., 1x1 pixel tasks in ARC, but potentially n x n pixels in more generalized visual reasoning), resizing or cropping—commonly used in standard vision tasks—is infeasible. VITARC reveals limitations in current ViT structures under these conditions and suggests necessary adapta-tions for such tasks.

Moreover, we believe that our insights into the importance of positional encodings for visual reason-ing tasks have implications beyond ARC, particularly for applications such as physical reasoning in vision generation tasks. In these contexts, accurate spatial relationships are equally critical, and our findings provide a foundation for further exploration of how Vision Transformers can be adapted to meet these challenges.

It is important to note that VITARC solves task-specific instances of ARC in a data-driven approach, treating each ARC task independently. This method does not fully solve ARC, which requires the ability to generalize across different tasks—a challenge that remains open for future research. How-ever, since the current state-of-the-art (SOTA) in ARC relies on LLM-based transduction models that handle tasks through supervised input-output transformations (arcprize, 2024), integrating the 2D inductive bias from ViTARC could provide an orthogonal benefit. This is especially relevant as prior studies indicate that the sequential nature of 1D methods in LLMs can limit ARC performance; for example, because the input grid is processed in raster order, LLMs experience a significant drop in success rates when horizontal movement/filling tasks are rotated 90 degrees (Xu et al., 2024).

In summary, this work highlights the importance of 2D positional information and object-based en-codings in abstract visual reasoning that leads to our novel contribution of the VITARC architecture. VITARC advances the application of Vision Transformers for pixel-level reasoning and suggests further avenues for improving generalization capabilities in models tackling visual reasoning tasks.

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

## A  VANILLA VIT FAILURE ANALYSIS

Figure 8: **Failure case of ViT-Vanilla with NLP `<pad>` tokens.** ViT-Vanilla with 2D padding and NLP `<pad>` tokens fails to account for the actual inner grid size, filling the entire $h_{max} \times w_{max}$ space. When the output is cropped to the true grid dimensions, the predictions within the valid region are correct, underscoring the importance of proper boundary handling.

## B  TRAINING DETAILS

This section provides a comprehensive overview of the training setup, including hyperparameters, hardware specifications, and other relevant details regarding the training process.

Our model consists of 3 layers with 8 attention heads and a hidden dimension of 128. The model was trained on various single-core GPU nodes, including P100, V100, and T4, with a batch size of 8 for 1 epoch. The typical training time per task ranges from 6 to 10 hours (wall clock).

The dataset was generated using Hodel's generators (Hodel, 2024), producing 1 million samples, which were then split into training, validation, and test sets with 998,000, 1,000, and 1,000 instances, respectively. The generation time varies between 3 and 12 hours, depending on the task. A fixed random seed (1230) was used for both dataset generation and model training to ensure reproducibility.

Due to computational resource constraints, the ablation study was performed on a randomly sampled subset of 100 tasks from the total 400, also selected using seed 1230.

## C  FULL RESULTS FOR TASK-SPECIFIC ACCURACIES

### C.1  MAIN MODELS ON FULL 400 TASKS

Table 1: Solved Test Instances (%) Across Models on all 400 tasks.

| Model | Solved Test Instances (%) | | | |
|---|---|---|---|---|
| | Mean | Med. | 25th Pctl. | 75th Pctl. |
| Baseline (ViT-Vanilla) | 17.68 | 3.20 | 0.10 | 22.85 |
| ViTARC-VT | 66.03 | 87.85 | 27.55 | 99.30 |
| ViTARC (Full Model) | 75.04 | 95.10 | 58.07 | 99.80 |

Table 2: Model accuracies across tasks (100/400)

| Task | ViT-Vanilla | ViTARC-VT | ViTARC | Task | ViT-Vanilla | ViTARC-VT | ViTARC |
|---|---|---|---|---|---|---|---|
| ce22a75a | 0.00 | 0.94 | 1.00 | 444801d8 | 0.00 | 0.98 | 1.00 |
| 1f876c06 | 0.00 | 0.99 | 1.00 | b27ca6d3 | 0.00 | 0.99 | 1.00 |
| 68b16354 | 0.00 | 0.99 | 1.00 | 2c608aff | 0.00 | 1.00 | 1.00 |
| d037b0a7 | 0.00 | 1.00 | 1.00 | 0ca9ddb6 | 0.00 | 1.00 | 1.00 |
| 543a7ed5 | 0.00 | 1.00 | 1.00 | 952a094c | 0.00 | 1.00 | 1.00 |
| af902bf9 | 0.00 | 1.00 | 1.00 | 49d1d64f | 0.00 | 1.00 | 1.00 |
| 0962bcdd | 0.00 | 1.00 | 1.00 | d364b489 | 0.00 | 1.00 | 1.00 |
| b60334d2 | 0.00 | 1.00 | 1.00 | a9f96cdd | 0.00 | 1.00 | 1.00 |
| 95990924 | 0.00 | 1.00 | 1.00 | 54d82841 | 0.00 | 0.80 | 0.99 |
| 25d487eb | 0.00 | 0.95 | 0.99 | 5c0a986e | 0.00 | 0.96 | 0.99 |
| d687bc17 | 0.00 | 0.97 | 0.99 | 363442ee | 0.00 | 0.98 | 0.99 |
| 6cdd2623 | 0.00 | 0.98 | 0.99 | db93a21d | 0.00 | 0.93 | 0.97 |
| 5168d44c | 0.00 | 0.94 | 0.97 | 3befdf3e | 0.00 | 0.97 | 0.97 |
| 22233c11 | 0.00 | 0.97 | 0.97 | 67a3c6ac | 0.00 | 1.00 | 0.97 |
| ae3edfdc | 0.00 | 0.72 | 0.96 | ded97339 | 0.00 | 0.92 | 0.96 |
| a2fd1cf0 | 0.00 | 0.95 | 0.96 | d4a91cb9 | 0.00 | 0.98 | 0.96 |
| d4f3cd78 | 0.00 | 0.99 | 0.96 | 6cf79266 | 0.00 | 0.96 | 0.95 |
| e98196ab | 0.00 | 0.99 | 0.95 | 56ff96f3 | 0.00 | 0.90 | 0.94 |
| 694f12f3 | 0.00 | 0.91 | 0.94 | 93b581b8 | 0.00 | 0.99 | 0.94 |
| 39e1d7f9 | 0.00 | 0.42 | 0.93 | 8403a5d5 | 0.00 | 1.00 | 0.93 |
| ecdecbb3 | 0.00 | 0.76 | 0.92 | 31aa019c | 0.00 | 0.82 | 0.90 |
| ec883f72 | 0.00 | 0.87 | 0.90 | 36fdfd69 | 0.00 | 0.75 | 0.89 |
| b7249182 | 0.00 | 0.74 | 0.88 | e9614598 | 0.00 | 0.86 | 0.88 |
| e76a88a6 | 0.00 | 0.00 | 0.87 | 3ac3eb23 | 0.00 | 0.71 | 0.87 |
| a64e4611 | 0.00 | 0.98 | 0.87 | 50846271 | 0.00 | 0.84 | 0.86 |
| 928ad970 | 0.00 | 0.97 | 0.86 | 40853293 | 0.00 | 0.99 | 0.86 |
| 6ecd11f4 | 0.00 | 0.00 | 0.84 | b527c5c6 | 0.00 | 0.66 | 0.84 |
| 1e0a9b12 | 0.00 | 0.69 | 0.84 | 7ddcd7ec | 0.00 | 0.75 | 0.84 |
| 2013d3e2 | 0.00 | 0.95 | 0.84 | e50d258f | 0.00 | 0.70 | 0.83 |
| 1caeab9d | 0.00 | 0.42 | 0.82 | 5ad4f10b | 0.00 | 0.62 | 0.82 |
| 98cf29f8 | 0.00 | 0.66 | 0.82 | 264363fd | 0.00 | 0.79 | 0.82 |
| 5521c0d9 | 0.00 | 0.75 | 0.79 | 0a938d79 | 0.00 | 0.86 | 0.78 |
| f8a8fe49 | 0.00 | 0.68 | 0.74 | a48eeaf7 | 0.00 | 0.76 | 0.73 |
| aba27056 | 0.00 | 0.59 | 0.70 | 2bcee788 | 0.00 | 0.64 | 0.70 |
| 47c1f68c | 0.00 | 0.45 | 0.68 | b548a754 | 0.00 | 0.95 | 0.68 |
| 890034e9 | 0.00 | 0.59 | 0.67 | 508bd3b6 | 0.00 | 0.66 | 0.64 |
| 6aa20dc0 | 0.00 | 0.33 | 0.63 | 2dd70a9a | 0.00 | 0.33 | 0.59 |
| 7c008303 | 0.00 | 0.48 | 0.58 | 6d58a25d | 0.00 | 0.33 | 0.56 |
| f8c80d96 | 0.00 | 0.13 | 0.55 | 6855a6e4 | 0.00 | 0.44 | 0.51 |
| 4093f84a | 0.00 | 0.31 | 0.49 | 90c28cc7 | 0.00 | 0.42 | 0.48 |
| db3e9e38 | 0.00 | 0.34 | 0.47 | 05f2a901 | 0.00 | 0.04 | 0.46 |
| 5c2c9af4 | 0.00 | 0.51 | 0.46 | d06dbe63 | 0.00 | 0.57 | 0.46 |
| 5daaa586 | 0.00 | 0.17 | 0.43 | f1cefba8 | 0.00 | 0.19 | 0.43 |
| 3906de3d | 0.00 | 0.28 | 0.42 | caa06a1f | 0.00 | 0.19 | 0.41 |
| 75b8110e | 0.00 | 0.62 | 0.40 | e8dc4411 | 0.00 | 0.28 | 0.39 |
| 8731374e | 0.00 | 0.22 | 0.38 | e48d4e1a | 0.00 | 0.30 | 0.38 |
| f35d900a | 0.00 | 0.65 | 0.38 | f15e1fac | 0.00 | 0.10 | 0.37 |
| 6e19193c | 0.00 | 0.12 | 0.37 | 3de23699 | 0.00 | 0.00 | 0.35 |
| 6b9890af | 0.00 | 0.00 | 0.35 | a78176bb | 0.00 | 0.26 | 0.32 |
| 1b60fb0c | 0.00 | 0.14 | 0.28 | e509e548 | 0.00 | 0.02 | 0.27 |

Table 3: Model accuracies across tasks (200/400)

| Task | ViT -Vanilla | ViTARC -VT | ViTARC | Task | ViT -Vanilla | ViTARC -VT | ViTARC |
|------|------|------|------|------|------|------|------|
| a1570a43 | 0.00 | 0.54 | 0.25 | 3e980e27 | 0.00 | 0.02 | 0.22 |
| 88a10436 | 0.00 | 0.00 | 0.20 | 9aec4887 | 0.00 | 0.02 | 0.19 |
| 7df24a62 | 0.00 | 0.10 | 0.19 | e21d9049 | 0.00 | 0.10 | 0.19 |
| 8a004b2b | 0.00 | 0.02 | 0.18 | 1f0c79e5 | 0.00 | 0.14 | 0.16 |
| 045e512c | 0.00 | 0.06 | 0.14 | ce602527 | 0.00 | 0.00 | 0.12 |
| b775ac94 | 0.00 | 0.03 | 0.12 | 8eb1be9a | 0.00 | 0.03 | 0.07 |
| fcb5c309 | 0.00 | 0.00 | 0.06 | a61ba2ce | 0.00 | 0.00 | 0.06 |
| 36d67576 | 0.00 | 0.04 | 0.06 | 846bdb03 | 0.00 | 0.00 | 0.05 |
| 234bbc79 | 0.00 | 0.00 | 0.05 | e40b9e2f | 0.00 | 0.02 | 0.05 |
| 57aa92db | 0.00 | 0.03 | 0.05 | 5117e062 | 0.00 | 0.00 | 0.04 |
| 8efcae92 | 0.00 | 0.00 | 0.04 | 72322fa7 | 0.00 | 0.02 | 0.04 |
| 623ea044 | 0.00 | 0.02 | 0.04 | 4938f0c2 | 0.00 | 0.07 | 0.04 |
| 3bd67248 | 0.00 | 0.08 | 0.04 | 48d8fb45 | 0.00 | 0.00 | 0.03 |
| a87f7484 | 0.00 | 0.00 | 0.03 | 447fd412 | 0.00 | 0.01 | 0.03 |
| e6721834 | 0.00 | 0.01 | 0.03 | 4c5c2cf0 | 0.00 | 0.08 | 0.03 |
| be94b721 | 0.00 | 0.00 | 0.02 | a8c38be5 | 0.00 | 0.00 | 0.02 |
| d07ae81c | 0.00 | 0.00 | 0.01 | 97a05b5b | 0.00 | 0.01 | 0.01 |
| 99b1bc43 | 0.00 | 0.00 | 0.00 | 137eaa0f | 0.00 | 0.00 | 0.00 |
| c8cbb738 | 0.00 | 0.00 | 0.00 | e5062a87 | 0.00 | 0.00 | 0.00 |
| 60b61512 | 0.01 | 0.83 | 1.00 | e8593010 | 0.01 | 0.83 | 1.00 |
| a79310a0 | 0.01 | 0.98 | 1.00 | d43fd935 | 0.01 | 0.98 | 1.00 |
| 253bf280 | 0.01 | 0.99 | 1.00 | dbc1a6ce | 0.01 | 1.00 | 1.00 |
| 4c4377d9 | 0.01 | 1.00 | 1.00 | 8be77c9e | 0.01 | 1.00 | 1.00 |
| 77fdfe62 | 0.01 | 1.00 | 1.00 | ed36ccf7 | 0.01 | 1.00 | 1.00 |
| 25ff71a9 | 0.01 | 1.00 | 1.00 | f5b8619d | 0.01 | 1.00 | 1.00 |
| dc1df850 | 0.01 | 1.00 | 1.00 | 10fcaaa3 | 0.01 | 0.99 | 0.99 |
| 178fcbfb | 0.01 | 1.00 | 0.99 | 3428a4f5 | 0.01 | 0.79 | 0.98 |
| 11852cab | 0.01 | 0.92 | 0.98 | 4612dd53 | 0.01 | 0.96 | 0.98 |
| fcc82909 | 0.01 | 0.96 | 0.97 | dc433765 | 0.01 | 0.91 | 0.96 |
| 39a8645d | 0.01 | 0.01 | 0.94 | 6fa7a44f | 0.01 | 1.00 | 0.94 |
| 834ec97d | 0.01 | 0.94 | 0.93 | 321b1fc6 | 0.01 | 0.55 | 0.92 |
| 4522001f | 0.01 | 0.22 | 0.88 | 88a62173 | 0.01 | 0.97 | 0.85 |
| d9f24cd1 | 0.01 | 0.67 | 0.74 | a65b410d | 0.01 | 0.69 | 0.74 |
| 9edfc990 | 0.01 | 0.33 | 0.48 | 6455b5f5 | 0.01 | 0.22 | 0.27 |
| 72ca375d | 0.01 | 0.01 | 0.14 | 3f7978a0 | 0.01 | 0.04 | 0.14 |
| f9012d9b | 0.01 | 0.02 | 0.02 | 0e206a2e | 0.01 | 0.02 | 0.02 |
| a8d7556c | 0.02 | 0.93 | 1.00 | 74dd1130 | 0.02 | 1.00 | 1.00 |
| d13f3404 | 0.02 | 1.00 | 1.00 | 6d0aefbc | 0.02 | 1.00 | 1.00 |
| c9e6f938 | 0.02 | 1.00 | 1.00 | 913fb3ed | 0.02 | 1.00 | 1.00 |
| 41e4d17e | 0.02 | 0.83 | 0.99 | 94f9d214 | 0.02 | 0.74 | 0.96 |
| 83302e8f | 0.02 | 0.75 | 0.94 | b94a9452 | 0.02 | 0.45 | 0.85 |
| 1f85a75f | 0.02 | 0.03 | 0.81 | b6afb2da | 0.02 | 1.00 | 0.77 |
| 6e82a1ae | 0.02 | 0.24 | 0.63 | 00d62c1b | 0.02 | 0.46 | 0.63 |
| 82819916 | 0.02 | 0.20 | 0.60 | 63613498 | 0.02 | 0.02 | 0.16 |
| 228f6490 | 0.02 | 0.03 | 0.06 | 09629e4f | 0.02 | 0.02 | 0.03 |
| 6d75e8bb | 0.03 | 0.99 | 1.00 | bc1d5164 | 0.03 | 1.00 | 1.00 |
| bdad9b1f | 0.03 | 1.00 | 1.00 | eb281b96 | 0.03 | 1.00 | 1.00 |
| e26a3af2 | 0.03 | 0.92 | 0.99 | 8d510a79 | 0.03 | 0.99 | 0.99 |
| f2829549 | 0.03 | 0.89 | 0.98 | 6430c8c4 | 0.03 | 0.89 | 0.98 |
| f25fbde4 | 0.03 | 0.02 | 0.96 | fafffa47 | 0.03 | 0.92 | 0.94 |

Table 4: Model accuracies across tasks (300/400)

| Task | ViT -Vanilla | ViTARC -VT | ViTARC | Task | ViT -Vanilla | ViTARC -VT | ViTARC |
|---|---|---|---|---|---|---|---|
| 6773b310 | 0.03 | 0.78 | 0.91 | a740d043 | 0.03 | 0.84 | 0.84 |
| 56dc2b01 | 0.03 | 0.43 | 0.58 | d2abd087 | 0.03 | 0.09 | 0.15 |
| 681b3aeb | 0.03 | 0.04 | 0.05 | 5bd6f4ac | 0.04 | 1.00 | 1.00 |
| 8d5021e8 | 0.04 | 1.00 | 1.00 | 3c9b0459 | 0.04 | 1.00 | 1.00 |
| 6150a2bd | 0.04 | 1.00 | 1.00 | 62c24649 | 0.04 | 1.00 | 0.99 |
| 3af2c5a8 | 0.04 | 1.00 | 0.99 | 1a07d186 | 0.04 | 0.84 | 0.98 |
| 855e0971 | 0.04 | 0.96 | 0.98 | 4258a5f9 | 0.04 | 0.97 | 0.98 |
| 3aa6fb7a | 0.04 | 1.00 | 0.98 | 6d0160f0 | 0.04 | 0.03 | 0.97 |
| 29ec7d0e | 0.04 | 0.62 | 0.83 | ae4f1146 | 0.04 | 0.14 | 0.67 |
| 760b3cac | 0.04 | 0.66 | 0.64 | 29623171 | 0.04 | 0.37 | 0.44 |
| 673ef223 | 0.04 | 0.30 | 0.26 | 2281f1f4 | 0.05 | 1.00 | 1.00 |
| cf98881b | 0.05 | 1.00 | 1.00 | ce4f8723 | 0.05 | 0.97 | 0.99 |
| 6c434453 | 0.05 | 0.93 | 0.96 | c1d99e64 | 0.05 | 0.99 | 0.95 |
| 2dc579da | 0.05 | 0.38 | 0.69 | c909285e | 0.05 | 0.20 | 0.58 |
| 73251a56 | 0.05 | 0.66 | 0.39 | 776ffc46 | 0.05 | 0.03 | 0.16 |
| 3345333e | 0.05 | 0.08 | 0.14 | beb8660c | 0.05 | 0.09 | 0.09 |
| 80af3007 | 0.06 | 0.98 | 1.00 | 7f4411dc | 0.06 | 0.95 | 0.99 |
| 32597951 | 0.06 | 0.98 | 0.99 | 7468f01a | 0.06 | 0.42 | 0.84 |
| 810b9b61 | 0.06 | 0.70 | 0.82 | a5313dff | 0.06 | 0.61 | 0.76 |
| ef135b50 | 0.07 | 0.99 | 1.00 | dae9d2b5 | 0.07 | 0.95 | 0.97 |
| 1c786137 | 0.07 | 0.05 | 0.75 | d8c310e9 | 0.07 | 0.72 | 0.74 |
| d22278a0 | 0.07 | 0.70 | 0.66 | d0f5fe59 | 0.08 | 0.09 | 1.00 |
| d5d6de2d | 0.08 | 0.98 | 1.00 | a416b8f3 | 0.08 | 1.00 | 1.00 |
| 1f642eb9 | 0.08 | 1.00 | 1.00 | c444b776 | 0.08 | 0.96 | 0.99 |
| cbded52d | 0.08 | 0.97 | 0.97 | 780d0b14 | 0.08 | 0.97 | 0.96 |
| 0b148d64 | 0.08 | 0.26 | 0.62 | b782dc8a | 0.08 | 0.30 | 0.28 |
| 9f236235 | 0.09 | 0.98 | 0.88 | 0dfd9992 | 0.09 | 0.67 | 0.84 |
| 7837ac64 | 0.09 | 0.82 | 0.82 | aabf363d | 0.09 | 0.12 | 0.73 |
| b8cdaf2b | 0.09 | 0.64 | 0.61 | a61f2674 | 0.10 | 0.75 | 0.84 |
| ce9e57f2 | 0.10 | 0.75 | 0.83 | 7b6016b9 | 0.10 | 0.65 | 0.80 |
| 0520fde7 | 0.11 | 1.00 | 1.00 | 496994bd | 0.11 | 1.00 | 0.97 |
| 150deff5 | 0.11 | 0.91 | 0.95 | 25d8a9c8 | 0.12 | 0.46 | 1.00 |
| 1b2d62fb | 0.12 | 0.99 | 1.00 | 1bfc4729 | 0.12 | 1.00 | 1.00 |
| 3618c87e | 0.12 | 0.98 | 0.99 | 90f3ed37 | 0.12 | 0.83 | 0.84 |
| 484b58aa | 0.12 | 0.54 | 0.66 | 662c240a | 0.12 | 0.77 | 0.42 |
| b2862040 | 0.12 | 0.30 | 0.39 | d90796e8 | 0.13 | 1.00 | 1.00 |
| 6a1e5592 | 0.13 | 0.18 | 0.22 | 42a50994 | 0.14 | 0.98 | 1.00 |
| 2bee17df | 0.14 | 0.99 | 1.00 | 67e8384a | 0.14 | 1.00 | 1.00 |
| 017c7c7b | 0.14 | 0.95 | 0.99 | a3325580 | 0.14 | 0.01 | 0.00 |
| ddf7fa4f | 0.15 | 0.78 | 0.95 | 23b5c85d | 0.16 | 0.03 | 0.24 |
| 05269061 | 0.16 | 0.12 | 0.22 | 22168020 | 0.17 | 1.00 | 1.00 |
| 23581191 | 0.17 | 0.92 | 0.96 | 53b68214 | 0.17 | 0.94 | 0.96 |
| 7e0986d6 | 0.18 | 0.97 | 1.00 | b190f7f5 | 0.18 | 0.97 | 0.98 |
| a3df8b1e | 0.18 | 0.14 | 0.22 | ea786f4a | 0.19 | 0.98 | 0.98 |
| 28bf18c6 | 0.19 | 0.07 | 0.81 | 3eda0437 | 0.19 | 0.68 | 0.69 |
| 22eb0ac0 | 0.20 | 0.96 | 1.00 | 3631a71a | 0.20 | 0.99 | 1.00 |
| aedd82e4 | 0.20 | 1.00 | 1.00 | 025d127b | 0.20 | 1.00 | 1.00 |
| 08ed6ac7 | 0.20 | 0.99 | 0.95 | 44d8ac46 | 0.20 | 0.59 | 0.86 |
| ff805c23 | 0.21 | 0.10 | 0.28 | e179c5f4 | 0.22 | 0.01 | 0.01 |
| 1cf80156 | 0.23 | 0.12 | 0.83 | f8ff0b80 | 0.23 | 0.33 | 0.65 |

Table 5: Model accuracies across tasks (400/400)

| Task | ViT -Vanilla | ViTARC -VT | ViTARC | Task | ViT -Vanilla | ViTARC -VT | ViTARC |
|------|------|------|------|------|------|------|------|
| 1fad071e | 0.23 | 0.24 | 0.59 | 9ecd008a | 0.23 | 0.16 | 0.24 |
| 67385a82 | 0.24 | 1.00 | 1.00 | 868de0fa | 0.24 | 1.00 | 1.00 |
| c9f8e694 | 0.24 | 1.00 | 1.00 | d6ad076f | 0.24 | 0.98 | 0.99 |
| dc0a314f | 0.24 | 0.14 | 0.24 | 27a28665 | 0.26 | 0.24 | 0.94 |
| 9af7a82c | 0.26 | 0.00 | 0.00 | 4290ef0e | 0.27 | 0.24 | 0.80 |
| 539a4f51 | 0.28 | 0.72 | 0.76 | cdecee7f | 0.28 | 0.04 | 0.11 |
| 99fa7670 | 0.29 | 1.00 | 1.00 | e73095fd | 0.29 | 0.98 | 0.99 |
| 9dfd6313 | 0.29 | 0.99 | 0.99 | b0c4d837 | 0.29 | 0.21 | 0.97 |
| 963e52fc | 0.30 | 1.00 | 1.00 | 941d9a10 | 0.30 | 0.98 | 0.99 |
| b230c067 | 0.30 | 0.44 | 0.46 | b9b7f026 | 0.31 | 0.37 | 1.00 |
| 06df4c85 | 0.31 | 1.00 | 1.00 | 67a423a3 | 0.32 | 1.00 | 0.99 |
| 54d9e175 | 0.33 | 1.00 | 1.00 | 28e73c20 | 0.33 | 1.00 | 0.98 |
| 6f8cd79b | 0.33 | 1.00 | 0.98 | ea32f347 | 0.34 | 0.65 | 0.71 |
| 97999447 | 0.35 | 1.00 | 1.00 | a85d4709 | 0.35 | 0.00 | 0.83 |
| a5f85a15 | 0.36 | 0.99 | 1.00 | c59eb873 | 0.36 | 1.00 | 1.00 |
| 7b7f7511 | 0.36 | 0.89 | 0.95 | d10ecb37 | 0.39 | 1.00 | 1.00 |
| d89b689b | 0.41 | 0.96 | 0.98 | de1cd16c | 0.41 | 0.37 | 0.97 |
| 29c11459 | 0.43 | 1.00 | 1.00 | 9172f3a0 | 0.43 | 1.00 | 1.00 |
| a68b268e | 0.44 | 1.00 | 1.00 | ba97ae07 | 0.44 | 1.00 | 1.00 |
| ff28f65a | 0.44 | 0.70 | 0.96 | 1190e5a7 | 0.44 | 0.81 | 0.91 |
| d406998b | 0.46 | 0.98 | 1.00 | ba26e723 | 0.47 | 1.00 | 1.00 |
| f25ffba3 | 0.50 | 0.99 | 1.00 | c3f564a4 | 0.52 | 0.94 | 1.00 |
| 2204b7a8 | 0.52 | 0.96 | 0.98 | 272f95fa | 0.54 | 1.00 | 1.00 |
| 91714a58 | 0.54 | 0.94 | 0.98 | 1e32b0e9 | 0.56 | 0.99 | 1.00 |
| d9fac9be | 0.57 | 0.68 | 0.97 | 44f52bb0 | 0.57 | 0.55 | 0.84 |
| d23f8c26 | 0.59 | 1.00 | 1.00 | b8825c91 | 0.60 | 0.99 | 0.99 |
| ac0a08a4 | 0.61 | 0.99 | 1.00 | bb43febb | 0.61 | 1.00 | 1.00 |
| c0f76784 | 0.61 | 1.00 | 1.00 | e9afcf9a | 0.62 | 1.00 | 0.98 |
| b91ae062 | 0.64 | 1.00 | 1.00 | cce03e0d | 0.64 | 1.00 | 1.00 |
| 007bbfb7 | 0.65 | 0.99 | 1.00 | 91413438 | 0.65 | 0.38 | 0.32 |
| c3e719e8 | 0.66 | 0.99 | 1.00 | e3497940 | 0.66 | 1.00 | 1.00 |
| d631b094 | 0.66 | 0.41 | 0.64 | 50cb2852 | 0.68 | 1.00 | 1.00 |
| 8e1813be | 0.70 | 0.99 | 1.00 | 9565186b | 0.74 | 0.96 | 1.00 |
| a699fb00 | 0.74 | 1.00 | 1.00 | 4347f46a | 0.76 | 1.00 | 0.99 |
| 469497ad | 0.76 | 0.92 | 0.95 | 239be575 | 0.76 | 0.74 | 0.82 |
| 8f2ea7aa | 0.81 | 0.23 | 0.98 | 5614dbcf | 0.82 | 1.00 | 1.00 |
| 9d9215db | 0.83 | 0.96 | 0.97 | 85c4e7cd | 0.84 | 0.99 | 0.90 |
| 8e5a5113 | 0.85 | 0.98 | 0.99 | 46442a0e | 0.86 | 1.00 | 1.00 |
| 7fe24cdd | 0.86 | 1.00 | 1.00 | 445eab21 | 0.86 | 0.92 | 0.96 |
| bd4472b8 | 0.89 | 0.49 | 0.58 | 3bdb4ada | 0.92 | 1.00 | 1.00 |
| bda2d7a6 | 0.94 | 0.98 | 1.00 | f76d97a5 | 0.94 | 1.00 | 1.00 |
| 2dee498d | 0.95 | 1.00 | 1.00 | 46f33fce | 0.96 | 1.00 | 1.00 |
| 746b3537 | 0.96 | 0.99 | 0.99 | eb5a1d5d | 0.97 | 1.00 | 1.00 |
| 0d3d703e | 0.98 | 1.00 | 1.00 | 5582e5ca | 0.98 | 0.95 | 0.99 |
| f8b3ba0a | 0.98 | 0.99 | 0.97 | feca6190 | 0.98 | 0.11 | 0.79 |
| 794b24be | 0.98 | 0.24 | 0.23 | d511f180 | 0.99 | 1.00 | 1.00 |
| b1948b0a | 0.99 | 1.00 | 1.00 | c8f0f002 | 0.99 | 1.00 | 1.00 |
| 995c5fa3 | 1.00 | 0.00 | 1.00 | 6e02f1e3 | 1.00 | 1.00 | 1.00 |
| bbc9ae5d | 1.00 | 1.00 | 1.00 | d4469b4b | 1.00 | 1.00 | 1.00 |
| 7447852a | 1.00 | 1.00 | 1.00 | 4be741c5 | 1.00 | 1.00 | 1.00 |

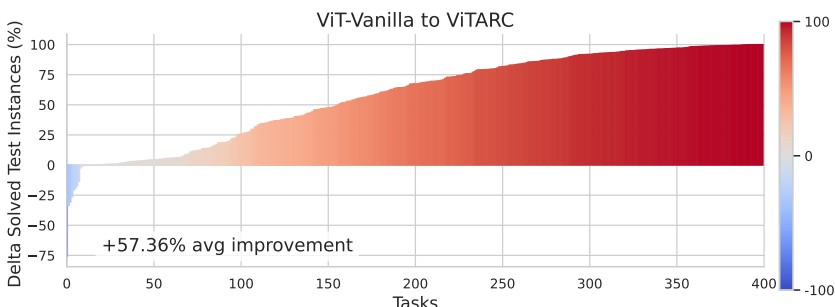

Figure 9: Improvement in percentage of solved test instances per task, from ViT-Vanilla to ViTARC.

## C.2  ABLATION MODELS ON SAMPLED 100 TASKS

Table 6: Solved test instances (%) across models on sampled 100 tasks and ablation of sub-steps. The Delta (Mean) column shows the change in the mean solved instances: the "Border Tokens" is compared to ViTARC-VT, while the three positional encoding ablations (PEmixer, 2D RPE, and OPE) are compared to ViTARC. Note that the numbers for ViT-Vanilla, ViTARC-VT, and ViTARC differ from the 400-task table as these are based on the 100-task subset.

| Model | Solved Test Instances (%) | | | | Delta (Mean) |
|---|---|---|---|---|---|
| | Mean | Median | 25th Pctl. | 75th Pctl. | |
| Baseline (ViT-Vanilla) | 15.98 | 3.65 | 0.10 | 15.90 | - |
| ViTARC-VT | 67.30 | 90.00 | 32.77 | 99.42 | base |
| - Border Tokens | 62.71 | 79.60 | 28.62 | 98.80 | -4.59 |
| ViTARC (Full Model) | 78.77 | 95.50 | 78.20 | 99.83 | base |
| - Positional Encoding Mixer (PEmixer) | 73.03 | 91.25 | 54.90 | 99.05 | -5.74 |
| - 2D Relative Positional Encoding (2D RPE) | 60.78 | 73.30 | 28.85 | 97.30 | -17.99 |
| - Object-based Positional Encoding (OPE) | 75.39 | 95.45 | 64.22 | 99.72 | -3.38 |

Table 7: Exact Match Scores for each task on 100 sampled tasks across different models and ablations.

| Task | ViT-Vanilla | ViTARC-VT | −BorderTokens | ViTARC | −PEmixer | −RPE | −OPE |
|---|---|---|---|---|---|---|---|
| 0ca9ddb6 | 0.00 | 1.00 | 1.00 | 1.00 | 0.27 | 1.00 | 1.00 |
| 543a7ed5 | 0.00 | 1.00 | 1.00 | 1.00 | 1.00 | 1.00 | 1.00 |
| 952a094c | 0.00 | 1.00 | 0.98 | 1.00 | 0.99 | 1.00 | 0.17 |
| 49d1d64f | 0.00 | 1.00 | 1.00 | 1.00 | 1.00 | 1.00 | 1.00 |
| 25d487eb | 0.00 | 0.95 | 0.99 | 0.99 | 0.08 | 0.95 | 0.37 |
| d687bc17 | 0.00 | 0.97 | 0.40 | 0.99 | 0.38 | 0.99 | 0.78 |
| 67a3c6ac | 0.00 | 1.00 | 0.84 | 0.97 | 0.99 | 1.00 | 1.00 |
| e98196ab | 0.00 | 0.99 | 0.96 | 0.95 | 0.92 | 1.00 | 0.09 |
| 8403a5d5 | 0.00 | 1.00 | 0.98 | 0.93 | 0.72 | 0.97 | 0.94 |
| 31aa019c | 0.00 | 0.82 | 0.69 | 0.90 | 0.89 | 0.99 | 0.81 |
| ec883f72 | 0.00 | 0.87 | 0.87 | 0.90 | 0.79 | 0.95 | 0.82 |
| b7249182 | 0.00 | 0.74 | 0.61 | 0.88 | 0.81 | 0.90 | 0.32 |
| e76a88a6 | 0.00 | 0.00 | 0.91 | 0.87 | 0.00 | 0.06 | 0.00 |
| 3ac3eb23 | 0.00 | 0.71 | 0.71 | 0.87 | 0.85 | 0.87 | 0.57 |
| a64e4611 | 0.00 | 0.98 | 0.97 | 0.87 | 0.90 | 0.99 | 0.99 |
| 40853293 | 0.00 | 0.99 | 0.92 | 0.86 | 0.98 | 0.98 | 0.96 |
| b527c5c6 | 0.00 | 0.66 | 0.74 | 0.84 | 0.56 | 0.76 | 0.53 |
| 2013d3e2 | 0.00 | 0.95 | 0.92 | 0.84 | 0.11 | 0.94 | 0.94 |
| 1caeab9d | 0.00 | 0.42 | 0.78 | 0.82 | 0.48 | 0.58 | 0.36 |
| 5521c0d9 | 0.00 | 0.75 | 0.69 | 0.79 | 0.76 | 0.80 | 0.71 |
| 6aa20dc0 | 0.00 | 0.33 | 0.52 | 0.63 | 0.38 | 0.51 | 0.23 |
| 2dd70a9a | 0.00 | 0.33 | 0.32 | 0.59 | 0.35 | 0.51 | 0.30 |
| 5c2c9af4 | 0.00 | 0.51 | 0.40 | 0.46 | 0.53 | 0.53 | 0.31 |
| 5daaa586 | 0.00 | 0.17 | 0.48 | 0.43 | 0.22 | 0.37 | 0.12 |
| 6e19193c | 0.00 | 0.12 | 0.18 | 0.37 | 0.29 | 0.08 | 0.08 |
| 1b60fb0c | 0.00 | 0.14 | 0.17 | 0.28 | 0.06 | 0.12 | 0.04 |
| 9aec4887 | 0.00 | 0.02 | 0.11 | 0.19 | 0.01 | 0.03 | 0.00 |
| 8a004b2b | 0.00 | 0.02 | 0.10 | 0.18 | 0.02 | 0.11 | 0.00 |
| 1f0c79e5 | 0.00 | 0.14 | 0.06 | 0.16 | 0.02 | 0.29 | 0.11 |
| a87f7484 | 0.00 | 0.00 | 0.01 | 0.03 | 0.00 | 0.14 | 0.00 |
| be94b721 | 0.00 | 0.00 | 0.02 | 0.02 | 0.01 | 0.00 | 0.00 |
| c8cbb738 | 0.00 | 0.00 | 0.01 | 0.00 | 0.00 | 0.01 | 0.00 |
| e5062a87 | 0.00 | 0.00 | 0.00 | 0.00 | 0.00 | 0.00 | 0.00 |
| d43fd935 | 0.01 | 0.98 | 0.98 | 1.00 | 0.97 | 0.99 | 0.99 |
| dbc1a6ce | 0.01 | 1.00 | 0.92 | 1.00 | 0.99 | 1.00 | 0.99 |
| dc1df850 | 0.01 | 1.00 | 1.00 | 1.00 | 1.00 | 1.00 | 1.00 |
| dc433765 | 0.01 | 0.91 | 0.92 | 0.96 | 0.68 | 0.97 | 0.94 |
| 39a8645d | 0.01 | 0.01 | 0.99 | 0.94 | 0.70 | 0.16 | 0.01 |
| 4522001f | 0.01 | 0.22 | 0.62 | 0.88 | 0.74 | 0.79 | 0.76 |
| 3f7978a0 | 0.01 | 0.04 | 0.12 | 0.14 | 0.06 | 0.11 | 0.01 |
| d13f3404 | 0.02 | 1.00 | 1.00 | 1.00 | 1.00 | 1.00 | 1.00 |
| 913fb3ed | 0.02 | 1.00 | 1.00 | 1.00 | 0.98 | 1.00 | 0.99 |
| 94f9d214 | 0.02 | 0.74 | 0.51 | 0.96 | 0.08 | 0.98 | 0.93 |
| 228f6490 | 0.02 | 0.03 | 0.06 | 0.06 | 0.04 | 0.04 | 0.02 |
| bdad9b1f | 0.03 | 1.00 | 1.00 | 1.00 | 1.00 | 1.00 | 1.0 |
| eb281b96 | 0.03 | 1.00 | 1.00 | 1.00 | 1.00 | 1.00 | 1.00 |
| 6430c8c4 | 0.03 | 0.89 | 0.53 | 0.98 | 0.43 | 0.99 | 0.96 |
| a740d043 | 0.03 | 0.84 | 0.65 | 0.84 | 0.64 | 0.82 | 0.46 |
| d2abd087 | 0.03 | 0.09 | 0.12 | 0.15 | 0.09 | 0.11 | 0.07 |
| 5bd6f4ac | 0.04 | 1.00 | 1.00 | 1.00 | 1.00 | 1.00 | 1.00 |

Table 8: Exact Match Scores for each task on 100 sampled tasks across different models and ablations.

| Task | ViT-Vanilla | ViTARC -VT | −BorderTokens | ViTARC | −PEmixer | −RPE | −OPE |
|------|-------------|------------|---------------|--------|----------|------|------|
| 8d5021e8 | 0.04 | 1.00 | 1.00 | 1.00 | 1.00 | 0.96 | 1.00 |
| 6150a2bd | 0.04 | 1.00 | 0.57 | 1.00 | 0.69 | 1.00 | 1.00 |
| 3af2c5a8 | 0.04 | 1.00 | 1.00 | 0.99 | 1.00 | 1.00 | 0.98 |
| 6d0160f0 | 0.04 | 0.03 | 0.93 | 0.97 | 0.02 | 0.98 | 0.56 |
| 29ec7d0e | 0.04 | 0.62 | 0.69 | 0.83 | 0.64 | 0.88 | 0.64 |
| 760b3cac | 0.04 | 0.66 | 0.60 | 0.64 | 0.11 | 0.78 | 0.47 |
| 6c434453 | 0.05 | 0.93 | 0.92 | 0.96 | 0.41 | 0.93 | 0.91 |
| c1d99e64 | 0.05 | 0.99 | 0.92 | 0.95 | 0.90 | 0.95 | 0.96 |
| 2dc579da | 0.05 | 0.38 | 0.55 | 0.69 | 0.43 | 0.71 | 0.16 |
| beb8660c | 0.05 | 0.09 | 0.06 | 0.09 | 0.08 | 0.13 | 0.06 |
| 7f4411dc | 0.06 | 0.95 | 0.98 | 0.99 | 0.90 | 1.00 | 0.97 |
| 32597951 | 0.06 | 0.98 | 0.98 | 0.99 | 0.97 | 1.00 | 0.99 |
| 1c786137 | 0.07 | 0.05 | 0.76 | 0.75 | 0.05 | 0.80 | 0.44 |
| d5d6de2d | 0.08 | 0.98 | 1.00 | 1.00 | 0.30 | 0.99 | 0.92 |
| 1f642eb9 | 0.08 | 1.00 | 0.92 | 1.00 | 0.89 | 1.00 | 0.98 |
| c444b776 | 0.08 | 0.96 | 0.98 | 0.99 | 0.93 | 0.98 | 0.82 |
| 0dfd9992 | 0.09 | 0.67 | 0.82 | 0.84 | 0.73 | 0.83 | 0.66 |
| 7837ac64 | 0.09 | 0.82 | 0.85 | 0.82 | 0.85 | 0.79 | 0.60 |
| a61f2674 | 0.10 | 0.75 | 0.71 | 0.84 | 0.84 | 0.86 | 0.54 |
| ce9e57f2 | 0.10 | 0.75 | 0.80 | 0.83 | 0.76 | 0.71 | 0.50 |
| b2862040 | 0.12 | 0.30 | 0.35 | 0.39 | 0.34 | 0.39 | 0.32 |
| d90796e8 | 0.13 | 1.00 | 1.00 | 1.00 | 0.85 | 1.00 | 1.00 |
| 42a50994 | 0.14 | 0.98 | 0.97 | 1.00 | 0.82 | 0.99 | 0.24 |
| 2bee17df | 0.14 | 0.99 | 1.00 | 1.00 | 0.01 | 1.00 | 0.98 |
| ddf7fa4f | 0.15 | 0.78 | 0.87 | 0.95 | 0.86 | 0.81 | 0.85 |
| 7e0986d6 | 0.18 | 0.97 | 1.00 | 1.00 | 1.00 | 0.99 | 0.99 |
| ea786f4a | 0.19 | 0.98 | 0.99 | 0.98 | 0.39 | 0.99 | 0.99 |
| 44d8ac46 | 0.20 | 0.59 | 0.70 | 0.86 | 0.79 | 0.74 | 0.63 |
| 868de0fa | 0.24 | 1.00 | 1.00 | 1.00 | 1.00 | 0.99 | 1.00 |
| dc0a314f | 0.24 | 0.14 | 0.24 | 0.24 | 0.28 | 0.35 | 0.01 |
| 9af7a82c | 0.26 | 0.00 | 0.00 | 0.00 | 0.00 | 0.00 | 0.00 |
| 99fa7670 | 0.29 | 1.00 | 1.00 | 1.00 | 0.97 | 1.00 | 1.00 |
| b0c4d837 | 0.29 | 0.21 | 0.91 | 0.97 | 0.21 | 0.93 | 0.13 |
| d89b689b | 0.41 | 0.96 | 0.97 | 0.98 | 0.93 | 0.98 | 0.38 |
| de1cd16c | 0.41 | 0.37 | 0.97 | 0.97 | 0.60 | 0.96 | 0.38 |
| a68b268e | 0.44 | 1.00 | 0.93 | 1.00 | 1.00 | 1.00 | 0.98 |
| d406998b | 0.46 | 0.98 | 1.00 | 1.00 | 0.39 | 1.00 | 0.73 |
| c3f564a4 | 0.52 | 0.94 | 0.94 | 1.00 | 0.88 | 1.00 | 0.92 |
| 44f52bb0 | 0.57 | 0.55 | 0.78 | 0.84 | 0.66 | 0.66 | 0.54 |
| ac0a08a4 | 0.61 | 0.99 | 0.98 | 1.00 | 1.00 | 1.00 | 1.00 |
| cce03e0d | 0.64 | 1.00 | 1.00 | 1.00 | 1.00 | 1.00 | 1.00 |
| 007bbfb7 | 0.65 | 0.99 | 1.00 | 1.00 | 0.84 | 1.00 | 1.00 |
| 91413438 | 0.65 | 0.38 | 0.34 | 0.32 | 0.33 | 0.32 | 0.92 |
| d631b094 | 0.66 | 0.41 | 0.43 | 0.64 | 0.64 | 0.73 | 0.05 |
| 445eab21 | 0.86 | 0.92 | 0.97 | 0.96 | 0.92 | 0.92 | 0.90 |
| 46f33fce | 0.96 | 1.00 | 1.00 | 1.00 | 0.84 | 1.00 | 1.00 |
| 5582e5ca | 0.98 | 0.95 | 1.00 | 0.99 | 0.98 | 0.97 | 0.96 |
| c8f0f002 | 0.99 | 1.00 | 1.00 | 1.00 | 1.00 | 1.00 | 1.00 |
| 995c5fa3 | 1.00 | 0.00 | 1.00 | 1.00 | 1.00 | 0.02 | 1.00 |
| 6e02f1e3 | 1.00 | 1.00 | 0.89 | 1.00 | 1.00 | 1.00 | 0.96 |

