# OpenReview forum: "Tackling the Abstraction and Reasoning Corpus with Vision Transformers: the Importance of 2D Representation, Positions, and Objects"
_ICLR.cc/2025/Conference — Submitted to ICLR 2025_

### Official Review · Reviewer_znVN · 2024-10-16

**Soundness:** 3
**Presentation:** 3
**Contribution:** 3
**Rating:** 8
**Confidence:** 4

**Summary:**

The paper studies the ARC benchmark, assessing the potential of ViTs for this task when trained with supervision on a single task at a time with plenty of (procedurally-generated) data (which is different from the original few-shot setting).

The paper first shows that standard ViTs fail. It then shows several modifications to the architecture that make them perform much better.

**Strengths:**

- The paper studies a popular architecture (ViTs), so it should be of broad interest.

- The paper shows a domain where ViTs fail dramatically. The task itself is not interesting (ARC in a new, supervised, large-data setting), but the finding is interesting because it points at intrinsic deficiencies of ViTs.

- Several modifications are described that improve the performance. It's kindof the opposite of an ablation study, but it does the job of demonstrating that novel methods imbue ViTs with novel capabilities.

**Weaknesses:**

I don't see significant flaws in this paper. Potential weaknesses:

- (W1) The proposed modifications (in particular the visual tokens) are quite straightforward. This can be seen a good thing. I am actually surprised that the absolute PE and padding (to handle images of different sizes or aspect ratios) have not been used before. Are the authors certain this hasn't been described in the existing literature?

- (W2) There is a valid argument that this paper solves a task that is extremely uninteresting in itself. It has no analogue in the real world and it completely defeats the purpose of the original ARC challenge (because of the supervised, large-data setting, focusing on a single task at a time). This paper makes absolutely no progress towards the goal of the ARC challenge. I still think that the findings are interesting for the reasons mentioned in my "Summary" above, i.e. that the proposed improvements give ViTs new abilities. The main issue now is that the contributions of this paper will only have value if/when the abilities are demonstrated to be useful for another task/setting.

**Questions:**

What do the authors think about adding a caveat in the paper about W2 above? (i.e. the fact that this makes zero progress on the ARC challenge, and that the benefits of the proposed methods still need to be demonstrated on a meaningful task)

I can't solve Task A (6e02f1e3) in Fig. 2. I guess other readers would benefit from an explanation? (I'm afraid this sortof makes a Vanilla ViT superhuman on this task!)

---

> ### Author Response · Authors · 2024-11-20
>
> We appreciate the reviewer’s encouraging comments.
>
> *Questions*
>
> **Adding caveat regarding the divergence from the ARC challenge, and that the real-world relevance is yet to be demonstrated.**
>
> We appreciate this point and will include this caveat in the updated version to clarify that ViTARC’s contributions are complementary to existing methods and that further testing on broader tasks would better demonstrate its utility (**ChangeLog #4.1**).
>
> ViTARC’s contributions are intended to complement current ARC solutions and support the challenge as a whole. Since the current SOTA in ARC relies on LLM-based transduction models that handle tasks through supervised input-output transformations [1], integrating the 2D inductive bias from ViTARC could provide an orthogonal benefit. Prior studies indicate that the sequential nature of 1D methods in LLMs can limit ARC performance; for example, because the input grid is processed in raster order, LLMs experience a significant drop in success rates when horizontal movement/filling tasks are rotated 90 degrees [2].
>
> As we will discuss in **Weakness #1**, the insights from ViTARC may also extend to reasoning tasks that require n x n pixel precision, with potential applications in physical-aware image or video generation in real-world scenarios.
>
>
> [1] Jack Cole and Mohamed Osman. Dataset-induced meta-learning (and othe tricks): Improving model efficiency on arc. https://lab42.global/community-model-efficiency/ ,2023.
>
> [2] Xu, Y., Li, W., Vaezipoor, P., Sanner, S., & Khalil, E. B. (2024). LLMs and the Abstraction and Reasoning Corpus: Successes, Failures, and the Importance of Object-based Representations. Transactions on Machine Learning Research (TMLR).
>
> **Explanation of Task A (6e02f1e3) in Fig. 2**
>
> Thank you for pointing this out. Task A in Fig. 2 follows a rule based on color count: if the input grid has two distinct colors, the output contains a grey diagonal from the top-left to the bottom-right. Conversely, if the input grid has three colors, the grey diagonal is from the top-right to the bottom-left. We agree that additional explanations or demonstrations for such tasks would be beneficial and have incorporated these clarifications in the updated version (**ChangeLog #4.2**).
>
> **Weaknesses #1: Straightforwardness of Modifications and Novelty in Existing Literature.**
>
> Treating each ARC task as an Abstract Visual Reasoning (AVR) task places us in a specific setup that combines reasoning, variable-sized grids, and a generative approach with pixel-level precision. Unlike standard vision tasks, we can’t use resizing or cropping to handle variable sizes, as ARC’s precision requirements make these methods infeasible.
>
> ARC tasks are also inherently more complex than benchmarks like RPM or odd-one-out (o3), which only require selecting a correct answer instead of generating a complete output. VQA, by contrast, involves vision input with NLP generation. Given these differences, to the best of our knowledge, there is no prior work addressing this same combination of requirements.
>
> Notably, the findings in ViTARC could be extended to support reasoning tasks requiring n x n pixel precision, including emerging discussions on physical reasoning in vision generation, where accurate spatial relationships are essential.
>
> **Weakness #2**
>
> Addressed in **Question #1**
>
> **ChangeLog 4.1**: Updated the conclusion to discuss broader contributions.
>
> **ChangeLog 4.2**: Added a description for Task A in Figure 2.

---

> > ### Comment · Reviewer_znVN · 2024-11-25
> > **Keeping my positive score**
> >
> > Thanks to the authors for the response. I pointed out several limitations of this work and its impact, and the authors agree to point them out more explicitly in the final version. Therefore I maintain my score. The rationale is that there are some people in the ICLR audience who will find the contributions of this paper interesting.

---

### Official Review · Reviewer_k3pK · 2024-10-21

**Soundness:** 3
**Presentation:** 2
**Contribution:** 2
**Rating:** 5
**Confidence:** 4

**Summary:**

The authors focus on a data-driven model to solve ARC. They first establish that vanilla ViTs fail on ARC despite being trained on a million examples. They then show that a 2D visual representation with ViT improves performance, and that object-based positional information further yields significant performance gains on ARC.

**Strengths:**

I appreciate that the authors explore the limits of a data-driven approach to ARC, as well as propose potential inductive biases to encode into a reasoning model for ARC. General priors for reasoning tasks are indeed important. The quantitative results compared to a naive ViT are promising for ARC.

**Weaknesses:**

1. Many of the architectural designs in the proposed model are made for solving ARC specifically. I believe ARC is a great intermediate proxy task for complex reasoning, but should not be an end goal in and of itself. With enough inductive biases, I believe that solving ARC with a million examples is reasonable, but is not particularly enlightening for the community. For example, 2D padding with <arc_pad> tokens and border tokens <arc endxgrid> that define grid conditions, etc, are very much defined for ARC itself.  Would like to see if this model generalizes to other reasoning tasks, for example Raven's progressive matrices, odd one out challenges, etc.
2. In addition, I'm not convinced that these are indeed the best inductive biases. For example, I believe that by using a "2D template" indicated by padding, border, and new-line tokens, the method is endowed with a priori knowledge of what the final grid shape should look like (and also, what the initial grid shape looks like). One core challenge of ARC is precisely that it needs to infer what the output dimensions are (how the shape transforms). Giving the model knowledge that one token should represent one block in ARC is a strong prior to be injecting.
3. The object-based positional encodings based on OpenCV's contour detections will struggle on more complex or different shapes. This “objectness” should be captured by the visual tokens implicitly.
4. No other baselines than ViT are explored, though there have been many works proposed for ARC & related reasoning tasks.

**Questions:**

1. Can you show that the model works on other complex reasoning tasks outside of ARC as mentioned in point 1 above?
2. Does the model work without strong priors, e.g., on input and output grid shape?
3. Can you add in more baselines trained on the same amount of data?

---

> ### Author Response · Authors · 2024-11-20
>
> We thank the reviewer for the thorough and thoughtful review.
>
> Before addressing your questions in detail, we would like to reiterate the main goal of this work. While vision transformers are being touted as a basic piece in vision and multimodal foundation models, we show that a vanilla vision transformer cannot solve ARC tasks even when trained on 1,000,000 examples. This finding indicates a clear representational deficiency in vision transformers when applied to visual reasoning. We diagnose this deficiency and address it in this work. Ultimately, the architecture and results are not limited to the ARC, as we will develop next. We focus on the ARC as a representative domain for visual reasoning with growing research interest from the community and reasonable computational resource needs that we can meet.
>
> *Weaknesses (part 1)*
>
> **Many of the architectural designs in the proposed model are made for solving ARC specifically.**
>
> We acknowledge that the naming of the padding tokens *<arc_pad>* and border tokens *<arc_endxgrid>* unintentionally suggests they are tailored specifically for the ARC domain. To clarify this, we will revise the terminology in the paper accordingly (**ChangeLog #3.1**).
>
> That said, the underlying implementation of 2D padding and border tokens is inherently generic and can be adapted to process any variable-sized images. While ARC demands pixel-level precision (1x1 patches), the approach has the potential to generalize to visual reasoning tasks involving n×n patches, where standard resizing or cropping techniques—typical in conventional vision tasks—are unsuitable.
>
> ViTARC stands out as the first approach to expose the representational deficiencies in current ViT structures under these constraints and to propose necessary adaptations for addressing such challenges. We will discuss the inductive bias issue and the potential to extend this work to other visual reasoning datasets in response to **Question #1**.
>
>
> **Strong inductive biases including a priori knowledge of what the final grid shape should look like, and also what the initial grid shape looks like.**
>
> We appreciate this concern, as the core challenge of ARC indeed involves inferring the output dimensions.
>
> The inductive biases we introduce are minimal and not specific to the ARC domain:
>
> a) The maximum grid size, a common practice in NLP tasks where padding requires a maximum length. Notably, most vision modeling imposes stricter constraints, such as fixed image sizes after resizing or cropping.
>
> b) The assumption that the maximum grid shape is rectangular, a typical prior in image processing.
>
> Importantly, these biases apply only to the **maximum** grids. We make no assumptions about the actual input or output grid shapes, which remain unbounded—even with the introduction of border tokens and 2D padding. This is especially true since ViTARC treats each pixel as a token.
>
> We hope these clarifications demonstrate that our injected biases are modest and consistent with standard practices in related fields. Regarding the comment, "Giving the model knowledge that one token should represent one block in ARC is a strong prior to be injecting," we are unclear about this interpretation. In ViTARC, each token represents one pixel so we assume the reviewer is referring to a pixel as a “block”?  If so, we see the pixel simply as the atomic representation of ARC tasks that is the most prior-free representation we can conceive.  If the reviewer was referring to a different meaning of “block”, we would appreciate it if they would consider clarifying their interpretation.
>
> **Object-Based Positional Encodings**
>
> We agree with the reviewer that, ideally, the model would learn to infer objectness implicitly through attention among pixels. However, as shown in our failure case analysis (Figure 6), the model sometimes requires a stronger signal to handle complex shapes, such as multi-colored or concave shapes. OPE serves as a demonstration of how architectural design can allow for the injection of external knowledge as “positional embeddings” within ViT models.
>
> Moreover, OPE works synergistically with PEmixer. If OPE’s signal is unreliable, the learned vector weighting can adapt to focus more on the original x and y coordinates. We will clarify this more clearly in the updated version (**ChangeLog #3.2**).

---

> ### Author Response · Authors · 2024-11-20
>
> *Weaknesses (part 2)*
>
> **No other baselines than ViT are explored, though there have been many works proposed for ARC & related reasoning tasks.**
>
> Our goal of the paper is to reveal limitations in current vision transformer architecture for performing abstract visual reasoning. Therefore, ViTARC is not yet a general ARC solver and cannot be compared directly with other ARC solvers. We aim to extend ViTARC to a general ARC solver in future work.
>
> Furthermore, AVR models currently rely on discriminative architectures rather than generative models. To our knowledge, there is no previous model positioned within the AVR + variable-sized grid + generative model regime. Therefore, we used the vanilla ViT setup as the baseline for this comparison. We would greatly appreciate any suggestions from the reviewer for relevant models or previous work to consider for comparison.
>
>
> *Questions*
>
> **Can you show that the model works on other complex reasoning tasks?**
>
> The diverse reasoning required across ARC tasks, ranging from spatial transformations to abstract rule discovery, already offers valuable insights into Abstract Visual Reasoning (AVR). A major strength of ViTARC lies in its dual capability as both a reasoning and generative model.
>
> While adapting ViTARC to other visual reasoning tasks is a promising future direction, it is beyond the scope of this paper. Notably, if benchmarks like RPM or odd-one-out were reframed as generative tasks, a significant portion would align with subsets of ARC.
>
> Additionally, our findings with ViTARC could also extend to reasoning tasks requiring n x n pixel precision, such as physical reasoning in vision generation, where spatial accuracy is equally critical. We have updated the paper to emphasize these broader contributions (**ChangeLog #3.2**).
>
> **Does the model work without strong priors, e.g., on input and output grid shape?**
>
> As previously described, we do not explicitly inject strong priors, such as predefined input or output grid shapes. However, if we were to remove the existing priors—such as 2D positional encoding, border tokens, and 2D padding—the setup would resemble a vanilla ViT more closely, and we expect the model’s performance would align similarly with it.
>
> **Additional Baselines Trained on the Same Amount of Data**
>
> Addressed in response to weakness 4.
>
> **ChangeLog 3.1**: Updated the terminology and description of vision tokens to ensure clarity and domain-agnostic applicability.
>
> **ChangeLog 3.2**: Updated the conclusion to discuss broader contributions.
>
> **ChangeLog 3.3**: Clarified OPE’s role in handling complex shapes, its synergy with PEmixer, and its novelty as a method for injecting external objectness knowledge.

---

> > ### Comment · Reviewer_k3pK · 2024-11-25
> > **Response to authors**
> >
> > Thanks to the authors for clarifying their method. I have updated my score accordingly, though I still note the limited evaluation on  ARC only, instead of other complex reasoning tasks (e.g., RPM and OOO as generative tasks) or physical reasoning tasks as the authors mentioned.

---

### Official Review · Reviewer_tj8k · 2024-11-02

**Soundness:** 2
**Presentation:** 3
**Contribution:** 3
**Rating:** 5
**Confidence:** 4

**Summary:**

This paper addresses the ARC benchmark from a vision perspective, and answers the question of what changes are required in current SOTA vision architectures to deal with the tasks in ARC.

The paper suggests that the bad results of vanilla ViTs on these tasks are due to their poor encoding of spatial and positional information, and proposes approaches to deal with this limitation.

Finally, the paper shows results that indicate that the proposed changes were useful for the performance on the ARC benchmark.

**Strengths:**

1 - The paper addresses an interesting question, which is: if the tasks in ARC are visual tasks, how can se use the current vision tools to deal with them?

2 - The reasoning behind every contribution in the paper is well explained. The paper is easy to follow.

3 - Related to the previous point, I particularly like the analysis in Figure 6. It helps understand what the model is (not) paying attention to.

4 - The results in the paper show a clear improvement with respect to the original ViT vanilla baseline, meaning the proposed contributions, overall, are helpful.

**Weaknesses:**

The paper has some weaknesses that I believe can be addressed, but also should be addressed.

**1 - Unclear that this is vision modeling**.

The main argument of the paper is that vision transformers should be improved to work on ARC. But the proposed changes make the model not be a vision model anymore. While the paper mentions that "Transformer-based LLM approaches convert the images into strings, which does not fully capture all relevant structural information”, this is not too different from what this paper does.
  - By construction, vanilla Transformers model inputs as sequences, but there are some aspects that make them more "vision" specific. Some of them are not included in the original ViT (e.g. local attention), and the others (e.g. patching) are removed in this paper (see next points). A non-Transformer-based architecture (e.g. a U-Net in a diffusion model) would make the connection to vision more clear. (I'm not suggesting such a model should be used, just exemplifying the point).
  - The pixels are encoded in a 1x1 grid, effectively making it a sequence.
  - There is an end-of-sequence token, just as there would be if the task was modeled as a sequence.
  - Even object positional encodings are added, abstracting away the low-level vision from the tasks.

An emphasis on vision is given in the paper more than once, e.g. "However, in vision tasks, especially those requiring detailed visual reasoning, spatial relationships often carry as much importance as, if not more than, the content of the tokens". I believe, however, that a lot of the learned lessons are not about vision, but about structured predictions. In general vision tasks, for example, the content of tokens *is* more important than the spatial relationships.


**2 - Unclear significance of contributions**.

Overall, the paper reads as a sequence of contributions the authors tried, one after another, and building on the previous one, without any global understanding of the problems. I believe the process to get to a solution should not be part of the paper. The paper should just present the final results and justify them. This makes the presentation confusing, for example when showing results in section 4, before going back to explaining some more technical contributions in section 5.

But the main problem is not about presentation. I believe there is some disconnect between the different contributions. Some of the latter ones should make the former ones unnecessary, and these are not ablated properly. The following are some examples:

  - First, learnable embeddings are removed in favor of positional embeddings. But then a PEmixer is necessary to learn position-specific embeddings. And also, RPE is required because APE encodes spatial changes very slowly. All of this is confusing. I believe the paper should directly start by explaining the final encoding and its reasoning, not present two different encoding techniques before the final one and show how they are not ideal.
  - The paper presents quantitative results showing that PEmixer and OPE, on top of ViTARC-VT actually *decrease* the performance of the model.
  - The padding is added at the beginning, but then it results in problems that need to be corrected. I address this in more detail in the next weakness.

**3 - It is unclear that all the padding contributions are necessary**.

Why are padding tokens necessary? The original formulation (sequential padding at the end), where the padding tokens are ignored by the attention should be the correct one (computationally it also makes more sense). The per-row padding (without being attended to) is effectively the same as the per-sequence padding, so I am not sure why it is being mentioned.

I understand that there is a problem about the output not understanding boundaries corrected. But the per-row EOS tokens should be enough to address this issue. For the model, it should be equally hard to predict an "end of row" token than it is to predict the current _<arc endxgrid>_ token. Would it be possible to ablate this? This is another example of weakness #2. The final solution should not include previous iterations of the solution, if these are not necessary. No (attended-to) padding would imply:
  - More efficient forward and backward passes.
  - No need for three different kinds of end-of-grid tokens. Only a single one would be enough.

**4 - No baselines or comparisons**.

There are no baselines or comparisons to other approaches, only to their own vanilla ViT.  Also, I could not find the paper's performance on the private set of the ARC benchmark, so it is hard to compare to SOTA approaches. Specially interesting comparisons would be to approaches that model ARC as a sequence using an LLM, as I believe the approaches are not very different (see weakness #1).

---

Other minor weaknesses:

- Figure 8 Appendix A seems to motivate the first main technical contributions of the paper. It should not be in the appendix.

- It is unclear why the PEMixer is helping. If every example in the task has different coordinates that are important, I don't understand why it would learn a global (across all samples) weighing. What would be the benefit of giving one position more weight than another one, if every example has different important positions?

- The paper could contain more clarifications and ablations. See "Questions" section.

**Questions:**

1 - Did you try using RoPE embeddings instead of RPE, in the second point of Section 5? I am curious about the differences.

2 - About RPE: the paper mentions there is an encoding for left and above, and another one for right and below. How are "above and to the right" patches encoded? Why not using an explicit "above left", "above right", "bottom left", "bottom right"? It seems like the approach is re-using the 1D sequential approach without taking into account that we are modeling 2D inputs.

3 - How do your architectural changes influence other vision tasks? E.g. classification, detection, VQA, etc. The SOTA ViT models are nowadays used for many tasks (at least as part of many tasks as vision encoders). It would be great if the changes proposed in the paper did not hurt performance in other tasks.

4 - Related to previous question, did you try starting from a pre-trained ViT?

5 - Did you try training jointly for all tasks? Adding a task ID as context for example.

6 - Did you notice any overfitting or underfitting on the final models? Any scaling properties with the data? On the final models, is 1M samples necessary, of 100k would be enough? Would the models perform better with even more samples?

7 - When generating pixels, during evaluation does the pixel value have to be exact? Is the prediction done as a classification in the RGB [256 x 256 x 256] space? Or is it a regression in the [0-1](x3) range? If it is a regression, how is the correctness evaluated?

---

> ### Author Response · Authors · 2024-11-20
>
> We thank the reviewer for the thorough and thoughtful review.
>
> *Questions:*
>
> **RoPE vs RPE?**
>
> We share this curiosity and agree that RoPE (or even 2DRoPE) could be an interesting future adaptation for ViTARC. However, it wasn’t included in the current experimental setup for the following reasons: a) We prioritized PE methods with more established compatibility since RoPE is not yet a standard approach for ViTs. b) We hypothesized that RoPE’s multiplicative nature doesn’t integrate smoothly with certain components in our design, such as PEmixer, which uses additive interactions. This creates challenges in adjusting the context-to-position ratio without compromising RoPE’s key property of position-difference sensitivity through dot product alignment. We will incorporate this discussion into our paper as future work.
>
> **Encoding Directions in RPE, why not use explicit encodings for all four diagonal directions?**
>
> Since the 2D image generation occurs in raster order, the pixels in the image can be related using only 2 directions without losing the 2D nature. This is reflected using our current 2-slope approach and we will clarify this in the updated paper (**ChangeLog #2.1**).
>
> However, we appreciate the reviewer’s suggestion of adding 2 additional slopes, which can potentially enhance precision by providing explicitly 2D representational capacity. We are conducting some experiments to test this four-slope approach and will aim to provide the results in a future update.
>
> **How do your architectural changes influence other vision tasks?**
>
> One of the main benefits of our proposed architectural changes is its enhanced focus on positional information, which we have found to be crucial for abstract visual reasoning tasks. This finding aligns with our intuition on vision-based tasks: spatial relations in higher dimensions carry more importance than in the 1D sequence setting. While we have not conducted experiments on other vision tasks, we hypothesize that this enhanced focus on positional information will be beneficial, especially in high-precision settings cases. We will incorporate this discussion into our paper as future works.
>
> **Starting from a Pre-trained ViT?**
>
> We agree that starting from a pre-trained ViT could be beneficial since fine-tuning a pre-trained (non-vision) transformer is currently state-of-the-art [1]. However, we did not conduct experiments using pre-trained ViTs, since the main goal of our study is to uncover and address the fundamental limitations within the current ViT setup which we believe is shown more prominently when studied in the simplest setting: a ViT trained for a single simple AVR task from scratch.
>
> [1] Jack Cole and Mohamed Osman. Dataset-induced meta-learning (and other tricks): Improving model efficiency on arc. https://lab42.global/community-model-efficiency/ ,2023.
>
> **Did you try training jointly for all tasks?**
>
> We are very interested in running this experiment, but are currently unable to do so given our computational resource limits.
>
> **Overfitting, Underfitting, and Data Scaling.**
>
> During development, our preliminary experiments suggested that 1M samples are indeed essential for the model size we used. For example, in ARC#007bbfb7 (https://kts.github.io/arc-viewer/page1/#007bbfb7), ViTARC-VT’s performance dropped significantly when using fewer samples: with 1M training samples, the model solved 99% of test instances, but reducing to 100k samples caused test performance to drop to 0.7%. As for additional data, while we haven’t tested beyond 1M samples, we have observed that training for more epochs improves performance. For instance, in ARC#1cf80156, increasing from 1 to 2 epochs improved ViTARC-VT’s performance from 9.6% to 17.5%. This suggests that increasing the training size could likely further enhance performance. We limited our experiments to 1 epoch due to the limitation in our computational resources.
>
> **Pixel Prediction Details**
>
> For the domain of ARC, each pixel takes only 1 of 9 possible colors. Therefore in ViTARC, each pixel corresponds to a single token in the vocabulary space and is handled as a standard transformer generation task, where an exact match is required.

---

> ### Author Response · Authors · 2024-11-20
>
> *Weaknesses (part 1)*
>
> **Unclear that this is Vision Modeling**
>
> We appreciate the reviewer’s concern that some of our adaptations to the ARC domain implemented for ViTARC make the model less "vision-specific." However, we argue that these adaptations are necessary for the ARC domain and do not shift our model away from vision modeling. Specifically:
>
> - ARC tasks demand pixel-level precision, which makes Conv-oriented setups (e.g., local attention or patching) less suitable. By using 1x1 patches, ViTARC ensures to capture every pixel detail. This adaptation doesn’t alter ViTARC’s core as a Vision Transformer as larger patch size can be implemented for tasks with greater tolerance (e.g., n x n pixel precision).
>
> - Unlike typical convolutional vision models where the image size and the number of patches are fixed, ViTARC must handle the generation of variable-sized grids. Since resizing or cropping is infeasible due to ARC's need for pixel-perfect precision, the End-of-Grid (EOS) token is a natural design choice. ViTARC-VT’s performance further demonstrates its importance.
>
> - We note that without positional encodings, a transformer inherently processes and outputs unordered sets. Thus, the 2D positional encodings, including OPE, explored in ViTARC reinforce our approach as being more vision-specific, not less as the reviewer suggests.
>
> We agree with the reviewer that in certain vision tasks, token content is more important than spatial relationships. However, in abstract visual reasoning tasks, such as those in ARC, spatial relationships sometimes hold primary importance. For instance, in tasks where same-color pixels move (e.g., ARC task #025d127b https://kts.github.io/arc-viewer/page1/#025d127b), the meaningful information lies in the positional shifts rather than in the token content itself.
>
> **Unclear Significance of Contributions**
>
> Thank you for raising concerns about disconnects between the different contributions. We’ll clarify each point below and update our paper accordingly.
>
> - PEmixer vs APE: PEmixer is primarily used to adjust the balance between positional and token information; it’s applied as an additional layer on top of APE.
>
> - RPE vs APE: As the reviewer pointed out, RPE was introduced to enhance spatial awareness that APE alone struggled with, as illustrated in the Figure 6 example. We acknowledge that the final encoding, which combines 2DAPE, 2DRPE, and OPE, may appear complex. We will clarify the progression and motivation behind each component in the updated version (**ChangeLog #2.2**).
>
> - PEmixer and OPE without RPE worsen performance: As discussed in section 5.1, this decrease in performance is due to OPE occupying space in the concatenated encoding and reducing the dimensionality available for APE. The inclusion of RPE recovers the positional signals at the attention level, thus improving the overall performance.
>
> We appreciate the reviewer’s concern about a lack of global understanding of the problems and the suggestions for restructuring the paper. Our paper introduces many improvements to the ViT architecture motivated by different failure analyses and aims to address very different challenges. Therefore, we structured our paper with the goal that readers can easily follow the progression of our various contributions, which we are grateful the reviewer acknowledges as part of the paper’s strength.
>
> **Necessity of Padding Contributions**
>
> We thank the reviewer for asking for additional clarification on the need for paddings. We address them in the following, and will update the paper accordingly:
>
> Sequential padding at the end where the padding tokens are ignored by the attention should be the correct one: While we can define 2D coordinates (x,y) for the encoder, where grid size is known in advance, the decoder lacks this spatial reference unless it operates on a fixed-size 2D grid. The <arc_pad> tokens serve as placeholders to support this fixed 2D template and therefore require attention. This setup was critical in enhancing performance from ViT to ViTARC-VT.
>
> Per-row EOS tokens should be enough and <arc endxgrid> are not needed: The need for border tokens arose after introducing 2D padding. Without them, the model would have to count tokens to determine boundaries, the precision of which quickly becomes an issue—especially in ARC tasks with dynamically defined output grid sizes (eg. task C in Figure 2). Additionally, inner grid borders differ from maximum grid boundaries, so separate tokens are used to prevent the model from having to learn this distinction implicitly. We will add these clarifications to the updated version (**ChangeLog #2.3**).

---

> ### Author Response · Authors · 2024-11-20
>
> *Weaknesses (part 2)*
>
> **No Baselines or Comparisons**
>
> We thank the reviewer for their concerns regarding a lack of comparison with other ARC models. We address each point below:
>
> - Performance on Private ARC Set: Our goal of the paper is to reveal limitations in current vision transformer architecture for performing abstract visual reasoning. Therefore, ViTARC is not yet a general ARC solver and cannot be applied directly to solve the overall ARC benchmark. We aim to extend ViTARC to a general ARC solver in future work, but resolving these critical learning deficiencies on within-task learning performance were our focus in this paper and a requisite stepping stone to future work on cross-task generalization with ViTs.
>
> - Baseline Comparisons: AVR models currently rely on discriminative architectures rather than generative models. To our knowledge, there is no previous model positioned within the AVR + variable-sized grid + generative model regime. Therefore, we used the vanilla ViT setup as the baseline for this comparison.
>
> - Comparison to Sequence-Based ARC Models Using LLMs:
> Each pixel in our setup is treated as a token, similar to many LLM-based ARC solver techniques. This design allows the vanilla ViT in our setup to function as a small-scale LLM **trained from scratch**, serving as a meaningful baseline to showcase the contributions of ViTARC's vision-specific architectural enhancements.
>
>     We acknowledge that comparing ViTARC with **pretrained** sequence-based LLMs could be an interesting direction for future work. However, such comparisons are beyond the scope of this paper, as their performance gains may stem from scaling laws rather than architectural advancements.
>
> *Minor Weaknesses:*
>
> **Placement of Figure 8**: We appreciate the reviewer’s recognition of Figure 8, however, we decided to move it to the Appendix as it only contributes partly as the motivation, and we did not want to further confuse the readers from the main motivation which is the example shown in Figure 4.
>
> **Purpose of PEmixer**: As noted earlier, the primary purpose of PEmixer is to control the ratio between contextual and positional information. We further opted for a vector PEmixer over a scalar for greater flexibility by allowing the model to learn when certain coordinates are more significant for a given task. For example, in ARC tasks with horizontal transformations, x-coordinates may hold greater importance.
>
> **ChangeLog 2.1**: Refined Section 5 on 2D-RPE to clarify the slope design.
>
> **ChangeLog 2.2**: Refined Sections 5 and 5.1 to better illustrate the different positional encodings (PEs) and provide a clearer discussion of the results.
>
> **ChangeLog 2.3**: Section 4, clarified the role of border tokens in 2D padding and dynamic grids.

---

> > ### Comment · Reviewer_tj8k · 2024-11-25
> > **I keep my rating**
> >
> > I appreciate the extensive rebuttal by the authors. They address the points that I raised in detail.
> >
> > However, I am still not convinced  that the contributions in this paper can be helpful for the community. The weaknesses that I raised in the initial review I believe still hold. A cleaner version of the approach and the paper, with more comparisons to existing methods, a more clear vision of what the contribution of the paper really is, and showing application to other tasks would greatly benefit the paper.

---

> ### Author Response · Authors · 2024-11-24
>
> **Follow-Up 2D RPE Experiment Results:**
>
> We conducted experiments to evaluate the reviewer’s suggestion of incorporating four diagonal directions into the relative positional encoding (RPE) design. These experiments were performed on a randomly selected subset of 50 tasks from the original 400 training tasks. The setup for the four diagonal directions was based on a 2D Manhattan distance approach, using the following slopes for the four directions:
> - Top-right: $ 1/2 $
> - Top-left: $ 1/2^{0.25} $
> - Down-right: $ 1/2^{0.5} $
> - Down-left: $ 1/2^{0.75} $
>
> Additionally, for comparison, we tested an alternative "4-direction" version that used slopes for only the up, down, left, and right directions. In this setup, pixels above and to the right received slope additions for their respective directions, divided by 2 to preserve normalization.
>
> ### Results Table
> Below is the performance summary for the three configurations:
>
> | Model (on 50 subset)      | Mean   | Median | 25th Pctl. | 75th Pctl. | Delta (Mean) |
> |---------------------------|--------|--------|------------|------------|--------------|
> | ViTARC (2 directions)     | 79.51  | 96.90  | 83.12      | 99.88      | base         |
> | 2D RPE - 4 diag directions| **79.74** | 94.20  | 77.72      | 99.80      | +0.23        |
> | 2D RPE - 4 directions     | 78.10  | 96.50  | 71.70      | 99.83      | -1.41        |
>
>
> ### Key Observations:
>
> 1. The 4-diagonal-direction RPE slightly outperformed the original 2-direction RPE in mean performance, supporting the reviewer's suggestion that explicitly modeling all four diagonal directions enhances precision.
>
> 2. While the performance gaps were not significant, both 2D Manhattan-based designs outperformed the cardinal-direction-based RPE (up, down, left, right). This indicates that incorporating diagonal relative distances provides additional spatial information, contributing positively to performance.
>
> These findings will be incorporated into future revisions and experiments as part of our continued refinement of RPE design.

---

### Official Review · Reviewer_NewP · 2024-11-09

**Soundness:** 3
**Presentation:** 3
**Contribution:** 2
**Rating:** 5
**Confidence:** 5

**Summary:**

This paper studies vision transformer in abstract visual reasoning ARC tasks which do not include any text or background knowledge, and focus purely on visual abstraction and pattern recognition. However, directly training a ViT with one million examples per task fails on most ARC tasks. Then, several techniques are proposed to improve model performances, including improved 2D positional encodings, and object-based positional encoding. This work highlights the importance of positional information for utilizing Vision Transformers in visual reasoning tasks.

**Strengths:**

1. The Abstract Visual Reasoning (AVR) tasks is interesting and important to study because it requires strong reasoning capability of ViTs. The paper also has very interesting findings, highlighting the importance of positional encodings in solving pure vision-based visual reasoning tasks.
2. This works provides very detailed model improvements they have tried to improve the performance, from ViT-Vanilla to ViTARC-VT and ViTARC. This is very useful to the communities to reproduce the experiments and improve further.
3. The final model ViTARC achieves very strong performance in most of the tasks, which is also a significant improvement over ViT-Vanilla.

**Weaknesses:**

1. The training/evaluation protocol is not clearly defined. The paper does not show clearly the generalization ability on unseen tasks. All the task they use for evaluation have some training examples during training. It would be very interesting to use some tasks pure for evaluation which are not seen during training.
2. Some of the key techniques used in this work are not new, like 2D (Relative) Positional Encoding, which have been discussed in the original ViT/Swin Transformer papers and plays a key role for performance improvement in this work. Though some new techniques are introduced in this work, like Positional Encoding Mixer (PEmixer) and Object-based Positional Encoding (OPE), the overall contribution and novelty is marginal.

**Questions:**

1. In figure 7, ViTARC-VT has very large variance in terms of performance. Any reason for it? Also what is the key technique leading to the significant performance improvement from ViT-Vanilla to ViTARC-VT? BorderTokens does not look to be important from this figure. Is this because of the 2D positional encodings in ViTARC-VT, instead of 1D positional encodings in ViT-Vanilla?
2. The Equation (12) is not quite clear. I did not find how to calculate the value of $r_{left}$ and $r_{right}$.
3. It is unclear how to tune the hyper-parameters $\alpha$ and $\beta$ in Equation (10), and no ablation studies are provided.

---

> ### Author Response · Authors · 2024-11-20
>
> We thank the reviewer for their detailed review.
>
> *Weaknesses*
>
> **Clarity on Training/Evaluation Protocol and Generalization Ability**
>
> Our main goal is to study the fundamental limitations of ViT in the simplest setting: a ViT trained for a single simple AVR task (ie. a single ARC task) from scratch. We observe that ViTs fail to learn on individual ARC tasks even when given 1,000,000 examples. Hence our primary objective in this paper is to understand why ViTs do not learn to generalize in-task and to resolve the sources of these learning deficiencies as we contributed in this paper. Addressing this critical gap on in-task ViT learning deficiencies is a requisite stepping stone for future extensions to cross-task generalization, an interesting direction we aim to explore in future work.
>
> **Novelty of Techniques and Limited Contribution**
>
>
> While we acknowledge that 2DAPE has been explored in prior ViT work, the effect of positional encodings has been largely underestimated due to the use of larger patch sizes, which results in fewer patches and reduces the influence of positional context per patch. For instance, the original ViT paper [1] reported minimal performance differences between 1DAPE, 2DAPE, and learned APE under these conditions. In contrast, we are the first to demonstrate that positional encodings become crucial in fine-grained reasoning settings, such as ARC with 1x1 patches, where positional context may dominate patch-level information—especially in tasks involving movement of same-color shapes (e.g. ARC#025d127b). This finding reveals a limitation in current vision reasoning approaches, as demonstrated by ViTARC’s improved performance.
>
> The necessity for enhanced 2D representations and an initial design similar to ViTARC has been independently highlighted by François Chollet, the creator of the ARC, in a recent Twitter post (Chollet, 2024) [2].
>
> Moreover, we believe this insight on the significance of positional encoding in visual reasoning tasks has implications beyond ARC, potentially informing fields such as physical reasoning in vision generation tasks. We have updated the paper to emphasize these broader contributions (**ChangeLog #1.1**).
>
> [1] Dosovitskiy. et al. (2021). An image is worth 16x16 words: Transformers for image recognition at scale. International Conference on Learning Representations.
>
> [2] Chollet, F. (2024). Twitter post. Retrieved from https://twitter.com/fchollet/status/1846263128801378616
>
> *Questions*
>
> **Variance in ViTARC-VT Performance (Figure 7)**
>
> The large variance in ViTARC-VT’s performance comes from the variety of difficulties across ARC tasks. Some tasks are almost always solved perfectly, while others tend to hover around 70% accuracy, which increases overall variance. Figure 2 (left side) shows this distribution more clearly, where a strong performance on many tasks is balanced by lower scores on a smaller group. While more training could possibly reduce this, our goal here was to compare different architectures under the same conditions rather than fine-tune for each task.
>
> **Key Technique for Performance Improvement**
>
> Regarding the main driver of improvement from ViT-Vanilla to ViTARC-VT, the key technique is indeed the use of 2D padding rather than BorderTokens. Applying 2DAPE directly is challenging with ARC’s variable grid sizes. While we could adjust 2DAPE to accommodate different input grid sizes in the encoder, the output grid would still be generated relative to unknown 2D positions, making it difficult to map accurately. By using 2D padding, we create a fixed generation schema—essentially a template that maintains consistent spatial alignment—which greatly improves performance. We appreciate the reviewer highlighting this, and we will update the experiment discussion in the paper to emphasize it (**ChangeLog #1.2**).
>
> **Clarification on Equation (12)**
>
> Thank you for this feedback. We will update the paper to clarify that $r_{left}$ and $r_{right}$ are fixed slopes following the original ALiBi setup, with adjusted starting values. We will update the paper to include the full equation and additional description for clarity (**ChangeLog #1.3**).
>
> **Tuning of Hyper-parameters α and β**
>
> α and β are not manually tuned but are learned parameters. We’ve updated the paper to clarify this point (**ChangeLog #1.4**).
>
>
>
>
>
> **ChangeLog 1.1**: Updated the conclusion to discuss broader contributions
>
> **ChangeLog 1.2**: Updated Section 4.1 to highlight 2D padding as the key driver of ViTARC-VT's performance boost and explain its role.
>
> **ChangeLog 1.3**: Clarified in Section 5 that $r_{left}$ and $r_{right}$​ follow a geometric sequence with different starting values, inspired by the ALiBi setup.
>
> **ChangeLog 1.4**: Updated section 5 PEmixer.

---

> ### Comment · Area_Chair_Luoa · 2024-11-26
> **[ACTION NEEDED] Respond to author rebuttal**
>
> Dear Reviewer,
>
> Now that the authors have posted their rebuttal, please take a moment and check whether your concerns were addressed. At your earliest convenience, please post a response and update your review, at a minimum acknowledging that you have read your rebuttal.
>
> Thank you,
> --Your AC

---

### Meta-Review · Area_Chair_Luoa · 2024-12-20

**Metareview:**

This is an interesting paper that designs a specific approach to tackle the ARC-AGI benchmark using object-centric priors and other task-specific components on top of a vision transformer architecture.

While the method is interesting and the results look encouraging, reviewers raised concerns about the experimental evaluation and the generality of the approach, as it is specifically designed for a single visual reasoning benchmark task. It would be valuable to test whether the method contributions provide value beyond ARC-AGI on other established visual reasoning tasks, such as Raven's Progressive Matrices etc.

**Additional Comments On Reviewer Discussion:**

Reviewer consensus was to reject the paper.

---

### Decision · Program_Chairs · 2025-01-22

Reject